# Improved inversion of aerosol components in the atmospheric column from remote sensing data

Ying Zhang[1], Zhengqiang Li[1], Yu Chen[2], Gerrit de Leeuw[1,3], Chi Zhang[1], Yisong Xie[1], Kaitao Li[1]

[1]Aerospace Information Research Institute, Chinese Academy of Sciences, Beijing 100101, China

[2]Public Meteorological Service Center, China Meteorological Administration, Beijing 100081, China

[3]Royal Netherlands Meteorological Institute (KNMI), R&D Satellite Observations, 3730AE De Bilt, The Netherlands

*Correspondence to*: Zhengqiang Li (lizq@radi.ac.cn)

**Abstract.** Knowledge on the composition of atmospheric aerosols is important for reducing the uncertainty of climate assessment. In this study, an improved algorithm is developed for the retrieval of atmospheric columnar aerosol components from optical remote sensing data. This is achieved by using the complex refractive index (CRI) of a multicomponent liquid system in the forward model and minimizing the differences with the observations. The aerosol components in this algorithm comprise five species, combining eight sub-components including black carbon (BC), water-soluble (WSOM) and water-insoluble organic matter (WIOM), ammonium nitrate (AN), sodium chloride (SC), dust-like (DU), and aerosol water content in the fine and coarse modes ($AW_f$ and $AW_c$). The calculation of the CRI in the multicomponent liquid system allows to separate the water-soluble components (AN, WSOM and $AW_f$) in the fine mode and the SC and $AW_c$ in the coarse mode. The uncertainty in the retrieval results is analyzed based on the simulation of typical models, showing that the complex refractive index obtained from instantaneous optical-physical inversion compares well with that obtained from chemical estimation. The algorithm was used to retrieve the columnar aerosol components over China using the ground-based remote sensing measurements from the Sun-sky radiometer Observation NETwork (SONET) in the period from 2010 to 2016. The results were used to analyze the regional distribution and interannual variation. The analysis shows that the atmospheric columnar DU component is dominant in the northern region of China, whereas the AW is higher in the southern coastal region. The SC component retrieved over the desert in northwest China originates from a paleo-marine source. The AN significantly decreased from 2011 to 2016, by 21.9 mg m$^{-2}$, which is inseparable from China's environmental control policies.

## 1. Introduction

Atmospheric aerosol consists of a suspension of solid and/or liquid particles in the air. The chemical composition and mixing state of the aerosol particles affect their optical characteristics, which in turn influence the energy budget of the Earth-atmosphere system and thus climate (Boucher et al., 2013).

To measure aerosol composition, many methods are used including online analysis in the field, sample analysis in the laboratory, remote sensing estimation, etc. Each technique provides information on the aerosol composition which may differ in content and detail. Because of fast observation and low cost, the application of remote sensing techniques to estimate aerosol

composition has developed rapidly since 2000. Satheesh et al. (1999; 2002a, b; 2005) established an algorithm for the inversion of aerosol components from remote sensing data based on the hypothesis of external mixing and assuming fixed size distributions for each component. But an external mixture usually cannot accurately describe the natural state of aerosols. Even if the particles are individually pure when first produced, numerous processes in the atmosphere will convert an external mixture to an internal mixture (Lesins et al., 2002). Therefore, internal mixing hypotheses are widely used and multiple approaches have been developed (e.g. Schuster et al., 2005, 2009, 2016; Arola et al., 2011; Li et al., 2013, 2019; Wang et al., 2013; van Beelen et al. 2014; Zhang et al., 2018). Schuster et al. (2005) determined the volume fraction of black carbon in an internal mixture with water and a soluble component by fitting the calculated complex refractive index to retrieved AERONET values at all four available wavelengths. In a follow-up study, Schuster et al. (2009) applied a similar procedure to determine the aerosol water fraction by fitting the real part of the refractive index of an internal mixture of water, soluble and insoluble species to observations by minimizing the cost function at all four wavelengths together. In this work the ratio of the dry volume fraction of insoluble to that of soluble aerosols was constrained by using a climatological value and the real refractive index which also prescribes the aerosol hygroscopicity. This constraint also provides a maximum insoluble fraction and the fraction of dust aerosol. Brown carbon was further estimated by Arola et al. (2011) due to the large change of its absorbing characteristic with wavelength for wavelengths smaller than 550 nm, but the dust component was ignored in this study. Aerosol bimodal characteristics were used by Schuster et al. (2016) to estimate the aerosol absorbing components including BC, brown carbon and hematite in the fine and coarse modes. This method was also embedded in the GRASP (Generalized Retrieval of Aerosol and Surface Properties; Dubovik et al., 2011)) system by Li et al. (2019) for application to POLDER/PARASOL observations. The above algorithms are aimed at retrieving absorbing aerosol components, such as BC, brown carbon and iron oxides, but have only simple treatment for scattering components, especially the host of multicomponent liquids.

van Beelen et al. (2014) introduced water-soluble organic matter (WSOM) in the inversion process based on the hygroscopicity of the OM mixture, but in this study water-insoluble organic matter (WIOM) was not accounted for. Some studies separated the OM based only on the spectral changes (Xie et al., 2017; Choi and Ghim, 2016) leading to large uncertainty in the results. Zhang et al. (2018) simultaneously retrieved the WSOM and WIOM components but ignored the error in the refractive index introduced by the aerosol volume averaging method applied to the multicomponent liquid system. For other non-absorbing components, the water content and inorganic components in the fine mode are identified by the difference in hygroscopic growth between organic and inorganic matter (Zhang et al., 2018; van Beelen et al.; 2014). In the coarse mode, sea salt is identified by the aerosol sphericity in the study of Xie et al. (2017) but this parameter is difficult to observe.

Although the retrieval of aerosol components by using remote sensing methods has been greatly developed, the application of hygroscopicity to identify the weak and non-absorbing components in a multicomponent liquid system remains difficult. In the current study, hygroscopicity is introduced to solve for the refractive index in a multicomponent liquid system. The results

are used in the algorithm to retrieve aerosol components from data obtained from the ground-based remote sensing network SONET (Li et al, 2018). The data and method are described in sections 2 and 3, respectively. The results for the aerosol components are presented and analyzed in section 4, and we conclude this study in section 5.

## 2 Measurements

### 2.1 Sun-Sky radiometer

The multiwavelength polarized sun-sky radiometer CE318-DP manufactured by Cimel Electronique in France, as an accurate instrument designed for long-term continuous observations in the field, can automatically measure solar and sky radiation. It consists of an optical head, a control box and a bi-axial stepping motor system. The optical head has two views: one for direct solar radiation with no focusing lens and the other for sky radiation with focusing lens. The internal optical system consists of a spectral and a polarizing filter to measure radiation in different wavebands with polarization directions. The 9 wavebands vary from the visible to the near-infrared (340, 380, 440, 500, 675, 870, 936, 1020, 1640 nm) with a full width at half maximum of 10 nm. All bands provide both radiation and polarization measurements, except the 936 nm band which only measures radiation to determine the columnar water vapor. These radiation and polarization measurements can provide sufficient information to calculate the columnar aerosol optical depth (AOD) and further retrieve the aerosol microphysical parameters.

### 2.2 SONET

The Sun-sky radiometer Observation NETwork (SONET) is a local observation network in China for ground-based remote sensing measurements of aerosol properties (Li et al., 2018). At present, there are 16 long-term observation sites in China, which are evenly distributed over north and south China, northwest China and the Tibetan Plateau (Figure 1). The longest time series is provided by the Beijing station, which was established in 2009. Five more stations joined in 2011 and 2012 and the network has been gradually growing to the current size. The geographical and topographical features of the long-term sites are diverse such as plateau, desert, hilly, plain and island, including three megacities, three islands and one plateau site (Table 1). SONET data provide sufficient variability, as regards length of time series, spatial coverage, climatic and topographic features and aerosol properties, for the analysis of atmospheric aerosol characteristics across China.

SONET provides continuous observations of direct sun and sky radiation measured using the multi-wavelength polarization sun-sky radiometer (CE318-DP), following the AERONET protocol (Li et al., 2018). Based on the inversion algorithm of Dubovik and King (2000) and Dubovik et al. (2000), the 440, 675, 870 and 1020 nm wavebands are used to retrieve more than 20 parameters describing the optical, physical and chemical global properties as column-integrated properties (Li et al., 2018), including particle volume size distribution (VSD), complex refractive index (CRI) and aerosol components. Using these data,

VSD and CRI sub-modal parameters of atmospheric aerosols are obtained using the modal decomposition method proposed by Zhang et al. (2017). The real parts of the CRI of the fine and coarse modes ($n_f$ and $n_c$, respectively) are spectrally independent, while the imaginary parts have spectral variation at 440 nm, so they are written as ($k_{f,440}$, $k_f$) & ($k_{c,440}$, $k_c$). Using these fine and coarse mode characteristics of the CRI, micro-physical properties of aerosols in each mode were analyzed (Li et al., 2019), but the aerosol chemical components were not determined.

**2.3 Meteorological data**

Meteorological data provide important supplementary information for the analysis and interpretation of the SONET-retrieved aerosol information. Hourly observations from surface meteorological stations were provided by the China Meteorological Administration (CMA). Only data from manned weather stations, which are maintained regularly, were used to ensure the best possible data quality. The CMA stations closest to each SONET site were selected and the meteorological data were collocated

in time with the SONET observations by linear interpolation between the nearest observations. Figure 2 shows the statistics of the relative humidity (RH) observations at each of the 16 sites. The highest mean RH occurs at the Sanya site, and the lowest value at the Lhasa site. Generally, the mean RH is relatively low at stations at northern latitudes, and often also at high altitudes. The standard deviations of wet (e.g. Sanya, Haikou) and dry sites (e.g. Lhasa, Kashgar) are smaller than at other sites.

**3. Methodology**

The aerosol components are determined by comparison of the aerosol microphysical properties calculated using a forward model with those retrieved from the SONET observations (Zhang et al., 2017). This is achieved by minimizing the iterative kernel function, i.e. the sum of the differences between the calculated and observed properties at each of the four wavelengths together, to find the optimum solution. The forward model includes three modules: the Maxwell Garnett effective medium approximation (Schuster et al., 2005) module to calculate aerosol internal mixing characteristics, an aerosol hygroscopic

growth module to solve the hygroscopicity of water-soluble components in a multicomponent liquid system, and an organic component dynamic constraint module to keep a reasonable ratio of organic matter.

**3.1 The aerosol component classification**

The aerosol component classification includes five principal species (black carbon (BC), organic matter (OM), inorganic salt (IS), aerosol water content (AW), dust-like (DU)). Three of these components are further sub-divided, i.e. organic matter is

sub-divided into water-soluble (WSOM) and water-insoluble organic matter (WIOM), inorganic salt consists of ammonium nitrate (AN) in the fine mode and sodium chloride (SC) in the coarse mode, and aerosol water content is the water content in the fine and in the coarse mode. Thus there are eight sub-components as illustrated in figure 3. All of these eight aerosol

components constitute a relatively complete system comparable to those used in chemical transport models.

The aerosol components are identified following three steps. The first step is the separation of the aerosol micro-physical properties (VSD and CRI) into those for the fine and coarse modes as summarized in supplementary S1. For the fine mode fraction, the water-insoluble and water-soluble components are identified using an empirical function (see section 2.2.2 in Zhang et al., 2018), which describes the ratio of the water-soluble to the water-insoluble volume fractions determined by RH, together with the parameterization of aerosol soluble volume fractions by Kandler and Schutz (2007). Then the subcomponents are separated into inclusion (BC and WIOM) and their environment (AN, $AW_f$ and WSOM) using their hygroscopic and optical absorption properties. It should be noted that the water-soluble property of aerosol components is not equivalent to hygroscopicity. Dicarboxylic acids represented by oxalic acid are dominant in the WSOM component but their hygroscopicity is extremely low (Ma et al., 2013; Drozd et al., 2014; Jing et al., 2016). Also other organic compounds in aerosols are less hygroscopic as shown in Zhang et al. (2018) (their figure 1). Hence, the OM components (WSOM and WIOM) are treated as non-hygroscopic components. For the coarse mode fraction, the refractive index of the mixture ($AW_c$ and SC) is determined by their hygroscopic growth factor. Dust and hydrate in the aerosol mixture are separated by the effective medium approximation.

In these processes, the hygroscopic growth is determined by the hygroscopicity parameter $\kappa$ and effective densities of the aerosol subcomponents, and the aerosol mixture refractive index is calculated by that of the subcomponents and the mixing state. Key parameters of the forward model and references are listed in table 2. We notice that the effective densities for OC and DU reported from different studies cover a wide range (Ganguly et al., 2009; McConnell et al., 2008; Wagner et al., 2012; Bond and Bergstrom, 2006) because they depend on the mixing ratios. In the current study the effective density of aerosol components is used from a widely cited study by van Beelen et al. (2014).

## 3.2 Complex refractive index in a multicomponent liquid system

The multiple water-soluble aerosol components together with the aerosol water content make up a liquid system, with increased complexity of the calculation of hygroscopic growth and complex refractive index. The $\kappa$-Köhler theory proposed by Petters and Kreidenweis (2007) can cope with the hygroscopicity of the multicomponent liquid system. In this theory, the water activity of aqueous atmospheric particulate matter can be represented by the functional form

$$\frac{1}{a_w} = 1 + \kappa \frac{V_s}{V_w}$$

(1)

where $V_s$ is the volume of the dry particulate matter and $V_w$ is the volume of the aerosol water content. The activity of water in solution ($a_w$) is close to the relative humidity (RH) due to lower curvature effect and can therefore be replaced with RH (Tang, 1996). The hygroscopicity parameter $\kappa$ is defined through its effect on the water activity of the solution. In equation (1), the

ratio of $V_s$ to $V_w$ can be further applied to the calculation of the volume fraction

$$\sum_i f_i = \frac{V_s}{V_s + V_w} = \frac{1 - a_w}{1 - (1 - \kappa)a_w}$$


(2)

where $f_i$ is the volume fraction of the $i$th component

$$f_i = \frac{V_i}{V_s + V_w}$$

(3)

where $V_i$ is the volume of the $i$th component.

In the multicomponent liquid system, the hygroscopicity parameter $\kappa$ is given by the simple mixing rule

$$\kappa = \sum_i f_{dry,i}\kappa_i$$

(4)

where $\kappa_i$ is the hygroscopicity parameter of the $i$th component obtained from the literature (table 2), and $f_{dry,\,i}$ is the dry

component volume fraction defined as

$$f_{dry,i} = \frac{V_i}{V_s}$$

160                                                                                                                    (5)

Using equation (2) for the relationship between the volume fraction and the hygroscopicity parameter, the complex refractive

index of the multi-component aerosol system can be derived using the Lorentz-Lorenz relation (Heller, 1965). Firstly, the

molar refractivity ($A_e$) at wavelength $\lambda$ can be calculated from the real part of the complex refractive index ($n_i$) and the volume

fraction of the individual components

$$A_e(\lambda) = \sum_i f_i A_i(\lambda)$$

(6)

Where $A_i$ is the molar refractivity of the $i$th component represented by


$$A_i(\lambda) = \frac{n_i^2(\lambda) - 1}{n_i^2(\lambda) + 2}$$

(7)

Then, the real and imaginary parts of the complex refractive index at wavelength $\lambda$ of the multi-component liquid system, $n_e(\lambda)$

and $k_e(\lambda)$, are obtained respectively by using the molar refractivity and the imaginary part of the complex refractive index of

the $i$th component ($k_i$).


$$n_e(\lambda) = \sqrt{\frac{1 + 2A_e(\lambda)}{1 - A_e(\lambda)}}$$

(8)

$$k_e(\lambda) = \sum_i f_i k_i(\lambda)$$

(9)

Equations (8) and (9) apply to the estimation of the complex refractive index of a multi-component liquid system with hygroscopic growth.

**3.3 Effective medium approximation**

To determine the complex refractive index of a particle, i.e. including both the multi-component liquid system and water-insoluble matter, the complex refractive index ($m = n - ik$) at wavelength $\lambda$ is expressed in terms of the permittivity, $\varepsilon(\lambda)$:

$$m(\lambda) = \sqrt{\frac{|\varepsilon(\lambda)| + Re(\varepsilon(\lambda))}{2}} + i\sqrt{\frac{|\varepsilon(\lambda)| - Re(\varepsilon(\lambda))}{2}}$$

(10)

The permittivity of the multi-component liquid system can then be calculated using equations (8) - (10). Considering the water-insoluble matter in a particle as inclusion and the water-soluble matter as the environment, the permittivity of the entire aerosol particle can be obtained by the Maxwell Garnett effective medium approximation (Schuster et al., 2005).

$$\varepsilon_{eff}(\lambda) = \varepsilon_e + 3\varepsilon_e \left[ \frac{\sum_j \dfrac{\varepsilon_j(\lambda) - \varepsilon_e(\lambda)}{\varepsilon_j(\lambda) + 2\varepsilon_e(\lambda)} f_j}{1 - \sum_j \dfrac{\varepsilon_j(\lambda) - \varepsilon_e(\lambda)}{\varepsilon_j(\lambda) + 2\varepsilon_e(\lambda)} f_j} \right]$$

(11)

where, $j$ is the number of water insoluble components and. $\varepsilon_j(\lambda)$ and $\varepsilon_e(\lambda)$ are the permittivities of the inclusion and its environment. The complex refractive index of the entire aerosol is estimated by aerosol component fraction using equation (10).

**3.4 Inversion procedure**

The flow chart for the inversion of the aerosol components is shown in figure 4. In the fine mode, the ratio of WS and WI
matter is estimated using RH as described in section 2.2.2 in Zhang et al. (2018). The initial value of the host refractive index and the extreme value for the BC component are set by the calculation modules of the complex refractive index in the multicomponent liquid system (see section 3.2) and the effective medium approximation (see section 3.3), respectively. In the loop to determine the BC component, two constraints are applied to separate BC from other components. The WSOM/WIOM ratio constraint was developed by Zhang et al. (2018) based on considerations published in the literature (Chalbot et al., 2016;
Bougiatioti et al., 2013; Wozniak et al., 2013; Mayol-Bracero et al., 2002; Krivácsy et al., 2001; Zappoli et al., 1999):

$$\begin{cases} f_{WSOM} \cong \alpha f_{WIOM} \\ \alpha = \dfrac{\beta \rho_{WSOM}^{-1}}{1 - \beta \rho_{WSOM}^{-1}} \qquad \beta \in [44\%, 77\%] \end{cases}$$

$$(12)$$

For more detail, see section 2.3.1 in Zhang et al. (2018). The volume normalization of the aerosol components in both the fine and coarse modes is used to constrain the volume fraction of the aerosol components to a reasonable range (similar as section

2.3.2 in Zhang et al., 2018)

$$\begin{cases} f_{fine} + f_{coarse} = 1.0 \\ f_{fine} = f_{BC} + f_{AN} + f_{WSOM} + f_{WIOM} + f_{AW_f} \\ f_{coarse} = f_{DU} + f_{SC} + f_{AW_c} \end{cases}$$

$$(13)$$

Then the inner loop of WSOM computes the CRIs of the fine mode at different BCs, and output the aerosol components of minimum $\chi^2$. The inversion procedure for the coarse mode is simpler than that for the fine mode. There is only a loop for DU

and the complex refractive index of the host can be directly calculated by equations (2) - (8) with only input of RH. The function Chi-squared ($\chi^2$) as an iterative kernel function is expressed in the sum of the differences between the complex refractive index estimated from the forward model ($m$) and the retrievals ($m_{rtrl}$), at multiple wavelengths:

$$\chi^2 = \sum_\lambda \frac{\left(m_{rtrl}(\lambda) - m(\lambda)\right)^2}{m_{rtrl}(\lambda)}$$

$$\lambda = 440, 675, 870 \text{ and } 1020 \text{ nm} \qquad (14)$$

The retrieval is completed when the value of $\chi^2$ reaches a minimum. The volume fractions of the aerosol components can be obtained by solving the above equations (10-12). The aerosol mass concentration in the atmospheric column is calculated using the volume and effective density of the aerosol components.

The retrieval algorithm described here is an improvement over that described in Zhang et al. (2018). In the previous algorithm, the WSOM component was added to the host, but it could only be considered as a non-hygroscopic component. The proportion

of solute and solution in the host mixture at different relative humidities should be measured in the laboratory, which limits the choice of aerosol components in the inversion process. Also, the real part of the CRI of the host was calculated by volume averaging, which can introduce a small error. The improved algorithm described here is more suitable for the calculation of the properties of a mixture of multiple water-soluble components as long as the hygroscopicity parameter is known, which is not only convenient to measure but also independent of particle size. The hygroscopicity parameter of WSOM can be varied

according to the choice of mixing components instead of changing the algorithm itself. Similarly, some other water-soluble components (e.g. sulfate) can be introduced into the inversion algorithm without laboratory measurements.

## 3.5 Uncertainty analysis

The uncertainty in the retrieval results was evaluated using synthetic data, both without and with input errors added. For the first case (without input errors), a set of complex refractive indices has been obtained by calculating a set of volume fractions of the aerosol components using the forward chemical model, which was used as input for the retrieval of the aerosol components without any noisy added. For the aerosol components, the volume fraction of BC was constrained between 0.0 to 3.0% with an interval of 0.5%, and corresponding dynamic ranges for the other components with intervals of 10%, in three ambient relative humidity conditions (40%, 60% and 80%). Figure 5 shows the comparison of the aerosol component volume fractions from forward modeling used as input, and their retrieved values. The volume fractions of the retrieved aerosol components reproduce the input values reasonably well. For the fine mode fraction, most data pairs are located close around the 1:1 line, with the mean absolute error (MAE) of the aerosol component volume fractions of 3.0%. In five samples the difference in the $AW_f$ is more than 20.0%, though the overall MAE for $AW_f$ is only 5.5%. In these five samples, the BC component is low and organic matter contributes substantially to the aerosol light absorption, resulting in underestimation of the $AW_f$ volume fraction at high RH and overestimation for moderate RH. WSOM is overall slightly overestimated and AN is underestimated by only a few percent. The correlation between the input and retrieved aerosol volume fractions in the coarse mode is even better than that in fine mode. The regression coefficient for all samples is 0.99, and the MAE is only 2.0%. These results show the very small uncertainty in the retrieved aerosol component volume fractions.

To further evaluate the inversion results, errors were added to the synthetic data. To this end, three typical pollution cases were chosen in which the main pollutants are water soluble, biomass burning and dust aerosols, respectively, further referred to as WS, BB and DU pollution types. Each type is described by the different aerosol size distribution and refractive index parameters derived from Zhang et al. (2017). These parameters are listed in the supplementary, table S1. Note that although the acronyms of the three pollution types are the same as the aerosol component names above, it does not mean that each type includes only one single aerosol component, as illustrated below.

Figure 6 shows the aerosol volume size distribution, complex refractive index and eight aerosol components in the WS, BB and DU types used in this exercise. For the size distribution, the highest volume concentrations occur in the fine mode of the WS and BB types, whereas for the DU type the coarse mode dominates. For the complex refractive index, significant absorption occurs in the fine mode fraction of the BB type, while relatively low absorption occurs in the other models. In the WS type, the mass fraction of $AW_f$ is close to 20% and for AN it is about 18%, significantly larger than for the other types. By comparison, the BC mass fraction in the BB type is close to 3%, and organic carbon is also high, with WSOM and WIOM mass fractions of 23% and 11%. In the DU type, the dust component is completely dominant, as expected, and the mass fractions of other components are less than 2%.

The three main sources of error in the model input parameters are the RH and the complex refractive index in the fine and

coarse modes. The uncertainties due to inversion errors of the modal refractive index were discussed in detail in Zhang et al. (2017) and are directly used here to estimate their effects on aerosol components. For RH, the observation error is about 5% (WMO, 2008), in this exercise a larger error (10%) is introduced to more rigorously assess the uncertainty in the estimated aerosol components. These typical uncertainties are listed in table S2. The total relative error (TRE), which is the propagated relative error calculated by the mean aerosol component error induced by the errors of sub-CRIs and RH in three pollution types, is used to assess the uncertainty in the aerosol composition inversion. As shown in table 3, the TRE of BC is 32.21%, less than other components in the fine mode, and the largest source of TRE is the imaginary part of the complex refractive index ($k_{f,440}$), with 25.68%. Compared with BC, the TRE of OM is larger (about 75%), primarily contributed by RH, followed by $n_f$. The uncertainty of the imaginary part impacts very little due to the low absorption of OM. The uncertainty of AN due to the imaginary part is low, but a very high uncertainty is caused by RH. Another component of IS is SC which usually occurs in the coarse mode. The large TRE of SC is contributed by the real part of the complex refractive index in the coarse mode ($n_c$), with 912.87%, leading to the largest TRE of IS. Affected by SC, the TRE of $AW_c$ is also large due to $n_c$, but the TRE of $AW_f$ is much smaller (50.05%). In the coarse mode, the TRE of DU is smallest in all of the aerosol components, only 15.79%, mainly caused by $n_c$. Overall, most of the uncertainties in the fine mode are from RH, and that in the coarse mode from the $n_c$. Fortunately, the RH observed by ground-based stations is accurate, with an error which is usually less than about 5% (WMO, 2008), which can significantly reduce the uncertainty in the retrieved aerosol scattering components. It should be noted that the uncertainties in table 3 are for single measurements. One important advantage of remote sensing is that multiple measurements can be made during a short period of time. Thus, the average uncertainty of the aerosol components can be effectively reduced by taking into account independent errors in each observation. In addition, the accuracy of the retrieved $n_c$ needs to be improved in order to deal with the aerosol component inversion.

## 4. Results

### 4.1 Aerosol component retrievals

The averaged mass fractions of the aerosol components measured at 16 SONET sites are presented in Figure 7. Each pie chart is marked with the site name, coordinates, observation period and BC fraction. The mass fractions are also listed in table S3. The pie charts show that the coarse mode mass fraction usually dominates at the northern and northwestern sites. The mass fraction of the dust component is significantly higher than that of others, with a fraction of more than 50% at the western (Lhasa, Zhangye, Kashgar, Minqin and Xi'an), Beijing, Harbin and Songshan sites, which is different from surface observations of chemical components (Zhang et al., 2012; Liu et al., 2014). This is because sun photometers provide data integrated over the whole atmospheric column and thus include the dust transport layer near 4 km (Proestakis et al., 2018), where dust concentrations may be substantial, whereas surface observations are local point measurements. The lowest dust

fractions are observed at southern sites, especially at the Guangzhou site, with a mass fraction of 31.5%. In contrast, the water content is dominant at southern sites in both the fine and coarse mode. The maximum AW ($AW_f$ and $AW_c$) fraction occurs at the Guangzhou site (28.7%), and the lowest mass fractions of 2.0% and 7.5% are observed at the Lhasa and Kashgar site, respectively. High $AW_f$ occurs in the cities of east-central China due to the higher occurrence of inorganic salts with larger hygroscopicity in the fine mode at these sites, whereas the dominant $AW_c$ in the western sites can be explained by the inorganic salt coating of larger particles in the dust source region (Rosenfeld et al., 2001). The IS fraction (AN and SC) gradually increases from north to south, which is consistent with the trend of the water content. The fraction of the AN sub-component is less than 7.0% at Lhasa, Zhangye, Kashgar and Minqin, whereas it is more than 20% at Chengdu, Guangzhou, Haikou and Sanya. At the Zhoushan site also a high AN fraction is observed, up to 17.1%. For the SC component, the maximum value occurs at the Kashgar (17.1%) site. The high SC fraction at the southeast coastal sites is readily ascribed to the influence of the ocean; the high SC fraction at the Kashgar site is due to the paleo-marine source of dust over the Taklimakan Desert (Huang et al., 2010). The WIOM component fraction is high in the central sites but relatively low in the southern coastal and northwest sites. For the WSOM component, the low value of less than 3% appears only at northwestern sites (Zhangye, Kashgar and Minqin). In the atmospheric column, the mass fraction of the BC component averaged over 16 sites is only 0.59%, lower than from near-surface in situ observations (usually 1% ~ 5%), which implies that the BC fraction may be reduced by the suspended layer with other components such as dust aerosol. Nevertheless, the unusually high mass fraction of BC in Shanghai could be due to observation uncertainty, also accompanied by the large error for aerosol component inversion.

The closure of the CRI between instantaneous optical-physical inversion and chemical estimation is examined by the data pair frequency. Figure 8 shows scatter density plots of the chemically estimated and sunphotometer-retrieved imaginary parts of the fine mode at 675 nm ($k_f$) and 440 nm ($k_{f,440}$) and the real parts of fine mode at 440 nm($n_f$). The points are colored by the number of data pairs (Retrieved, Estimated), which are sorted according to ordered pairs in 0.0005 intervals for the imaginary parts of CRI and 0.001 intervals for the real parts. The data pairs of $k_f$ are closely concentrated around the 1:1 line, although a slight underestimation is observed with 94.3% of the estimated values lower than the retrieved values; only 5.3% of the data pairs have a relatively large absolute error (AE > 0.01). The mean bias is not large (-0.003), and the mean absolute value is equal to the mean absolute error (MAE = 0.003). There are two reasons for this slight underestimation in chemical estimation. On the one hand, the imaginary part of the refractive index of BC is much larger than for the other components due to its strong absorption. Thus, the inversion of the BC concentration is very sensitive to the estimation of the refractive index. As shown in table 3, although the TRE of BC is the lowest, the errors caused by $k_f$ and $k_{f,440}$ are larger than for any other component. On the other hand, $k_f$ is not only affected by BC in the inversion process, but also affected by organic components (WSOM & WIOM) with spectral absorption characteristics. Therefore, in most cases, $k_f$ is underestimated in chemical estimation and $k_{f,440}$ is overestimated (Bias = 0.007). The mean relative error (RE) is 27.1%, and 62.8% of the data points are below the average

relative error line. This indicates that most inversion results have good optical closure. For the closure of the real part of the fine mode, the data pairs of $n_f$ are also concentrated around the identity line, although 76.5% of the $n_f$ is above the identity line. Underestimation occurs mainly when $n_f$ is larger than 1.56, because the only component with the real part of the CRI larger than 1.56 is BC, but its concentration is mainly determined by the imaginary part. The bias of the estimated $n_f$ (Bias = 0.009) is larger than that of $k_f$ due to the fact that the value and the range of $n_f$ are larger than that of $k_f$.

In addition, the comparison of aerosol components with that from Zhang et al. (2018) is given in the supplementary (S3). The figure S1 and S2 show the algorithm in this study shows a positive effect on AN and $AW_f$, although there are few validation points.

**4.2 Seasonal variation**

The seasonal variation of the aerosol component mass concentrations, averaged over 15 stations (Lhasa is not used due to lack of adequate seasonal data) and all available years, is shown as box-whisker plots in Figure 9. The top and bottom edges of each box represent the top and bottom quartiles (Q3 and Q1), and the corresponding whiskers are the outliers (Q3+1.5IQR and Q1-1.5IQR, IQR is interquartile range). The mean value is indicated by a plus sign (+), and the median value by a short line inside the box (–). Figure 9 shows that the DU component exhibits an obvious seasonality. The DU mass concentration is very high in the spring and the mean value reaches up to 332.9 mg m$^{-2}$ due to dust transport from the northwest of China. With the weakening of dust transport and the increase of moisture, the DU fraction decreases in other seasons, with a mean value of around 240.0 mg m$^{-2}$. Although the DU concentration is lower in the summer than in other seasons, it is still relatively high near the dust source area, which results in a large difference between the upper and lower quartiles. In contrast, the AN mass concentration mean value peaks in the summer (76.8 mg m$^{-2}$), whereas a minimum occurs in the spring (47.7 mg m$^{-2}$). It is worth noting that although the mean value in the winter is not high (51.1 mg m$^{-2}$), the interval between the upper and lower quartiles of AN is the smallest in the winter. The minimum value of AN (17.9 mg m$^{-2}$) is higher than in other seasons (4.1 mg m$^{-2}$ in spring, 9.5 mg m$^{-2}$ in summer, and 11.1 mg m$^{-2}$ in autumn). The seasonal variation of the water content is slightly different from that of inorganic salts. The low values of mean $AW_f$ occur in the spring, while $AW_c$ is significantly lower in the winter (21.0 mg m$^{-2}$) than in other seasons. The difference between the upper and lower quartiles of $AW_f$ in the summer is larger than in other seasons indicating that in the summer the aerosol at some sites has a low hygroscopicity. The OM mass concentration is slightly higher in the winter than that in other seasons probably due to the occurrence of haze pollution in the winter, with mean concentrations of the WIOM and WSOM fractions of 22.3 and 38.8 mg m$^{-2}$, respectively. In the summer, the OM concentration is only about two thirds of that in the winter. The median value of the BC mass concentration is higher in the winter (3.0 mg m$^{-2}$), which can be related to heating in northern China. Low concentrations of BC in the other seasons are mainly due to the influence of frequent dust events in the spring and high aerosol hygroscopic growth in the summer.

Similar to AN, the SC concentration peaks in the summer and has a minimum in the winter, due to the influence of the Asian monsoon. The median values in these two seasons are respectively 41.6 and 19.6 mg m$^{-2}$.

The seasonal variation of the main aerosol components in the fine mode is discussed on a regional basis (Figure 10). BC concentrations in typical northern regions are higher than in southern regions, because of emissions due to winter heating only in the north. Other BC sources are vehicle emissions and biomass burning. Adverse meteorological conditions in winter result in the accumulation of BC in the atmosphere resulting in high BC values in both the north and the south. The highest BC mass concentrations in the northern region in the winter is 4.3 mg m$^{-2}$. OM is one of the dominant components in the fine mode, with sources similar to those of BC. The impact of biomass burning in the winter and spring over south China (Chen et al., 2017) is significant, leading to OM concentrations of more than 50.0 mg m$^{-2}$. In the northern region, much biomass burning occurs in the autumn (Wang et al., 2020). With the influence of heating, the OM level in the north can reach up to 80.1 mg m$^{-2}$. Therefore, the OM mass concentration in the northern region is only low in the summer (50.8 mg m$^{-2}$). AN is usually formed by secondary reactions of gaseous precursors in complex air pollution areas. In both the northern and the southern region, AN mass concentration is larger in the summer than in other seasons, and the seasonal variation in the southern region is significantly smaller than that in the north. The mean AN mass concentration in the southern region is 8.7 mg m$^{-2}$ higher than that in the northern region. This suggests that more AN is produced by secondary reactions in the humid climate in the south than in the northern region.

## 4.3 Interannual variation

Figure 11 shows the interannual variations of the aerosol component mass concentrations in the atmospheric column from 2010-2016. The 16 SONET sites have been established in succession, so the number of available observations increased year by year with the longest time series from the Beijing site (see also Table 1). The annual mean mass concentrations shown in Figure 11 are averages over all sites, i.e. the number of sites was not accounted for and, in particular in the earlier years (2010-2011), the annual mean may thus be representative for one (Beijing) or a few sites. Therefore, the annual means for each site available has been plotted as well. Figure 11 shows that the annual mean mass concentrations of most of the aerosol components in the fine mode increased in most of the first years and then decreased. Influenced by China's environmental control policies, the mean AN decreased significantly from 72.4 mg m$^{-2}$ in 2011 to 50.5 mg m$^{-2}$ in 2016, i.e. a reduction by 21.9 mg m$^{-2}$. The yearly mass concentrations of AN at most sites also follow a downward trend, and AN in the southeastern coastal sites are significantly higher than that in the northwestern sites. In contrast, the mean BC mass concentration shows a peak (3.9 mg m$^{-2}$) in 2011, drops in 2012 to the lowest value during the whole period (2.3 mg m$^{-2}$), then increases somewhat to a second peak (2.7 mg m$^{-2}$) in 2013. After a decrease in 2014, BC climbed to 2.8 mg m$^{-2}$ in 2016. In the southeastern coastal and northwestern sites, BC concentrations were relatively low. The unusually high values at Shanghai in 2016 may be due to

observational errors. Similar to BC, $AW_f$ also experienced a small fluctuation after a significant decline in 2012. The $AW_f$ in aerosol measured at the southern sites are higher than that at other sites. The fine mode WIOM and WSOM components show different behaviour. WIOM reached a peak in 2013, with the peak value of 32.3 mg m$^{-2}$, and then showed a significant decline after 2013. WSOM also reached a peak concentration of 35.8 mg m$^{-2}$ in 2013, which is 2 mg m$^{-2}$ lower than the peak in 2016, and overall the concentrations increased. These results suggest that the policy of air pollution control in China is effective in controlling inorganic salts and WIOM aerosols, while WSOM still needs to be further controlled. The concentrations of the coarse mode aerosol components fluctuate somewhat during the observation period, with a slight peak in 2013. The concentration of each component in the coarse mode at the northwestern sites is higher than that at other sites, which can be related to the high fraction of large particles. Due to the large influence of geographical factors on the coarse mode aerosol components, DU in 2010 (only Beijing site) was significantly larger than in other years. Since 2014, the mean DU mass concentration has increased, while a downward tendency is observed in the $AW_c$ and SC concentrations since 2013. Coarse mode aerosols usually derive from natural sources, and their variations can be associated with changes in the meteorological conditions.

## 5. Conclusions and discussions

The accurate measurement of atmospheric aerosol components plays an important role in reducing the uncertainty of climate assessment. In the current study, we updated the refractive index calculation in a multi-component liquid system and improved the component inversion algorithm of Zhang et al. (2018) to retrieve atmospheric columnar aerosol components including black carbon (BC), organic matter (WSOM/WIOM), inorganic salt (AN & SC), dust-like (DU) and water content in the fine and coarse modes ($AW_f$ & $AW_c$). This algorithm was applied to data from the SONET sun photometer network, and the regional distribution and interannual variation of atmospheric aerosol components in China were analyzed for the period from 2010 to 2016. The results show that the dust-like component is dominant in northern China, but the aerosol water content ($AW_f$ & $AW_c$) is dominant in the southern coastal region. The inorganic salt (AN) in the fine mode has a significant seasonal variation, with a mass concentration of 76.8 mg m$^{-2}$ in the summer which is significantly higher than that in other seasons. Meanwhile, the AN concentrations have significantly decreased from 2011 to 2016, which is inseparable from China's environmental control policies. However, the slight increase in WSOM and BC is still noteworthy.

As the aerosol concentrations in the atmospheric column obtained from the inversion of remote sensing data are different from those measured by in situ measurements near the surface, such as on-line aerosol mass spectrometers, the validation of the retrieval results is difficult. Proestakis et al. (2018) used data from the Cloud-Aerosol Lidar with Orthogonal Polarization (CALIOP) on the CALIPSO satellite to analyze the distribution of mineral dust over China and the results show the higher concentration of the DU component in the atmospheric column over northern China. Similarly, Huang et al. (2010) provided

a basis for the high SC content at the Kashgar station due to the paleo-marine source. However, for the direct comparison of our retrievals with independent data, airborne measurements of the vertical distribution of atmospheric aerosol components are needed (Kahn et al., 2017). In future research, we will design a verification experiment to comprehensively evaluate the results from our inversion method.

The method presented can be used not only for ground-based sun-sky photometer measurements, but also for other remote sensing instruments (e.g. lidar), and even for satellite remote sensing in the future. Meanwhile, as long as measurements of multi-wavelength extinction coefficients and aerosol particle size distributions are available, the inversion of atmospheric particulate matter composition can also be performed using comprehensive observations with multiple instruments near the surface. Therefore, this method can be widely used in low-cost and wide-area measurements in the future, providing a possibility for obtaining the global distribution of aerosol composition.

*Data availability.* The aerosol component data used in this study can be requested from the corresponding author (lizq@radi.ac.cn).

*Author contributions.* ZL conceived and designed the study. YC collected and processed the meteorological data. KL and YX collected the remote sensing data. CZ collected the DEM data and draw the map. YZ improved the aerosol component method and performed the inversions. YZ analyzed the spatiotemporal trends of aerosol component concentrations. YZ and GL prepared the paper with contributions from all coauthors.

*Acknowledgments.* This work was supported by the National Key R&D Program of China (grant 2016YFE0201400) and the National Natural Science Fund of China (41601386).

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

**Table 1**. SONET sites (name, location and geographical aspects) and meteorological stations used in this study.

| | | SONET Site | | | | Meteorological Station | | | | Geo feature | Geo region |
| Name | Abbr | Lon (°) | Lat (°) | Alt (m) | Obs Period | No.[*] | Lon (°) | Lat (°) | Alt (m) | | |
|---|---|---|---|---|---|---|---|---|---|---|---|
| Lhasa | LS | 91.2 | 29.6 | 3678 | 2016.03-2016.05 | 55591 | 91.1 | 29.7 | 3649 | Plateau | Qinghai-Tibetan |
| Kashgar | KS | 75.9 | 39.5 | 1320 | 2013.09-2016.11 | 51709 | 76.0 | 39.5 | 1289 | Desert | Northwest |
| Zhangye | ZY | 100.3 | 38.8 | 1364 | 2012.08-2016.10 | 52652 | 100.4 | 38.9 | 1483 | Gobi & desert | |
| Minqin | MQ | 103.0 | 38.6 | 1589 | 2012.02-2016.10 | 52681 | 103.1 | 38.6 | 1368 | Desert & hill | |
| Xi'an | XA | 108.9 | 34.2 | 389 | 2012.05-2016.11 | 57039 | 108.9 | 34.2 | 433 | Half mountain, half plain | |
| Beijing | BJ | 116.3 | 40.0 | 59 | 2009.12-2016.11 | 54399 | 116.3 | 40.0 | 46 | Hill (megacity) | Northern |
| Harbin | HrB | 126.6 | 45.7 | 223 | 2013.12-2016.11 | 50953 | 126.8 | 45.8 | 118 | Plain | |
| Songshan | SS | 113.1 | 34.5 | 475 | 2013.12-2016.11 | 57084 | 113.1 | 34.5 | 1178 | Mountain & hill | |
| Nanjing | NJ | 119.0 | 32.1 | 52 | 2013.01-2016.07 | 58238 | 118.8 | 32.0 | 35 | Plain & hill | Southern |
| Shanghai | SH | 121.5 | 31.3 | 84 | 2013.03-2016.04 | 58362 | 121.5 | 31.4 | 6 | Alluvial plain (megacity) | |
| Hefei | HF | 117.2 | 31.9 | 36 | 2013.11-2016.11 | 58321 | 117.2 | 31.9 | 27 | Hill | |
| Zhoushan | ZS | 122.1 | 29.9 | 29 | 2012.02-2016.11 | 58477 | 122.1 | 30.0 | 36 | Islands | |
| Chengdu | CD | 104.0 | 30.6 | 510 | 2013.06-2016.07 | 56276 | 103.8 | 30.4 | 461 | Basin | |
| Guangzhou | GZ | 113.4 | 23.1 | 28 | 2011.10-2016.11 | 59287 | 113.3 | 23.2 | 41 | Mountain, hill & plain (megacity) | |
| Haikou | HK | 110.3 | 20.0 | 22 | 2014.03-2016.03 | 59758 | 110.3 | 20.0 | 64 | Island | |
| Sanya | SY | 109.4 | 18.3 | 29 | 2014.09-2016.11 | 59948 | 109.5 | 18.2 | 419 | Island | |

[*] "No." is the meteorological station number.

**Table 2.** Growth factor derived hygroscopicity parameter ($\kappa$), complex refractive indexes ($m = n - ik$) at four wavelengths and effective density ($\rho$) of model components. Real and imaginary parts at four standard AERONET aerosol product wavelengths (440, 675, 870 and 1020 nm) are considered.

| Component | | Growth factor derived $\kappa$ | Real Part | | | | Imaginary Part | | $\rho$ (g cm$^{-3}$) |
|---|---|---|---|---|---|---|---|---|---|
| | | | $n_{440}$ | $n_{675}$ | $n_{870}$ | $n_{1020}$ | $k_{440}$ | $k_{675\sim1020}$ | |
| OM | WIOM | 0.000 | 1.530[c] | 1.530 | 1.530 | 1.530 | 0.035[d] | 0.001 | 1.547[i] |
| | WSOM | 0.000[a] | 1.530[c] | 1.530 | 1.530 | 1.530 | 0.006[d] | 0.000 | |
| AN | | 0.547[b] | 1.559[e] | 1.553 | 1.550 | 1.548 | 0.000[e] | 0.000 | 1.760[i] |
| BC | | 0.000 | 1.950[f] | 1.950 | 1.950 | 1.950 | 0.790[f] | 0.790 | 1.800[i] |
| AW | | 0.000 | 1.337[e] | 1.332 | 1.330 | 1.328 | 0.000[g] | 0.000 | 1.000[i] |
| DU | | 0.000 | 1.534[g] | 1.534 | 1.534 | 1.534 | 0.002[h] | 0.001 | 2.650[i] |
| SC | | 1.120[a] | 1.562[h] | 1.541 | 1.534 | 1.530 | 0.000[i] | 0.000 | 2.165[i] |

[a] Petters and Kreidenweis, 2007; [b] Kreidenweis et al., 2008; [c] Sun et al., 2007; [d] Chen and Bond, 2010; [e] Schuster et al., 2005; [f] Bond and Bergstrom, 2006; [g] Koven and Fung, 2006; [h] Toon et al., 1976; [i] van Beelen et al., 2014.

**Table 3.** Estimated total relative errors (TRE) of aerosol component mass fractions in the three aerosol models used to evaluate the aerosol component classification inversion algorithm.

| Aerosol components | | RH | Fine mode | | | Coarse mode | | | TRE* | |
|---|---|---|---|---|---|---|---|---|---|---|
| | | | $n_f$ | $k_{f,440}$ | $k_f$ | $n_c$ | $k_{c,440}$ | $k_c$ | | |
| BC | | 5.74% | 0.59% | 25.68% | 18.57% | 0.00% | 0.00% | 0.00% | 32.21% | |
| OM | WIOM | 75.82% | 4.55% | 5.28% | 1.08% | 0.00% | 0.00% | 0.00% | 76.15% | 74.73% |
| | WSOM | 51.60% | 51.92% | 3.44% | 2.11% | 0.00% | 0.00% | 0.00% | 73.31% | |
| IS | AN | 207.00% | 60.86% | 7.07% | 6.04% | 0.00% | 0.00% | 0.00% | 215.96% | 564.42% |
| | SC | 25.71% | 0.00% | 0.00% | 0.00% | 912.51% | 2.16% | 1.23% | 912.87% | |
| AW | AW$_f$ | 49.77% | 3.80% | 3.71% | 0.00% | 0.00% | 0.00% | 0.00% | 50.05% | 481.32% |
| | AW$_c$ | 8.95% | 0.00% | 0.00% | 0.00% | 912.55% | 2.10% | 1.17% | 912.60% | |
| DU | | 0.34% | 0.00% | 0.00% | 0.00% | 15.78% | 0.04% | 0.02% | 15.79% | |

*TRE $= \sqrt{\sum_1^7 \bar{x}_i^2}$, where $\bar{x}$ represents the mean error of aerosol components from three aerosol types. The RH is given input error of $\pm10\%$ and the inversion errors of sub-CRIs is from Zhang et al., 2017 listed in Table S2.


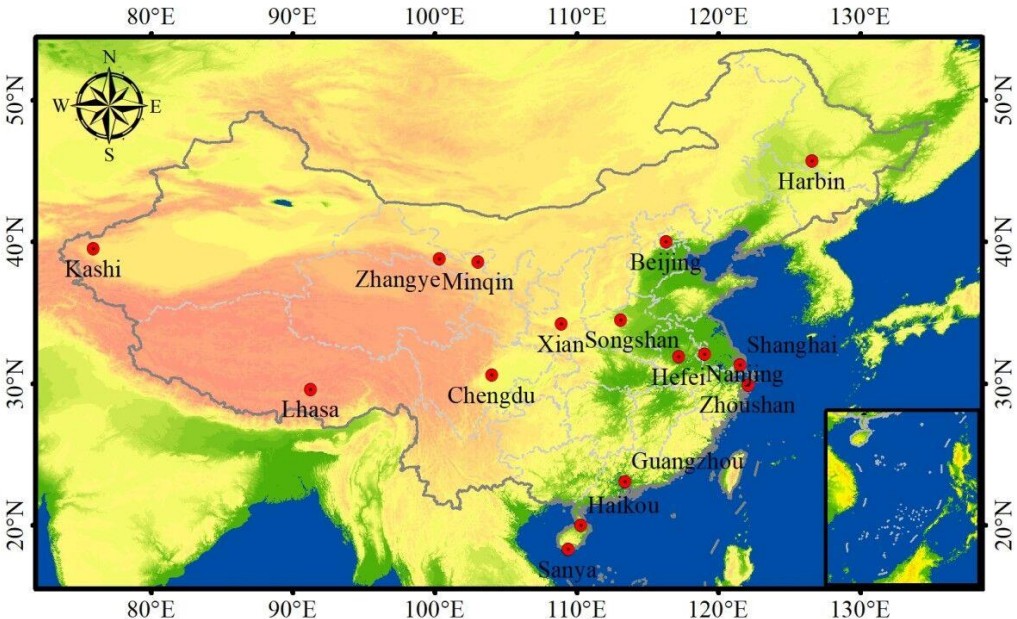

**Figure 1**. Locations of the 16 Sun-sky radiometer observation network (SONET) sites projected on the elevation map of China.

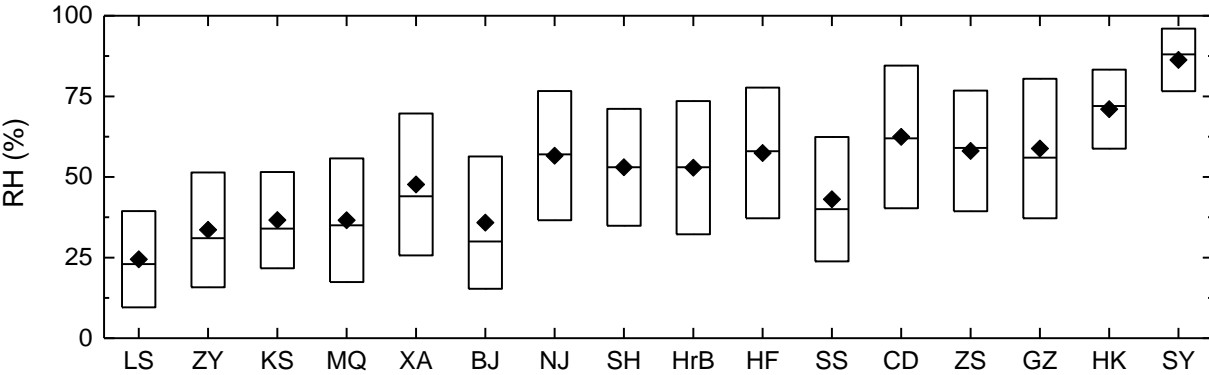

**Figure 2**. Boxplots of the relative humidities observed near each of the SONET sites. The observation periods for each site are shown in table 1. The line and the diamond represent the median and mean values, respectively, and the box shows the standard deviation (1 σ).

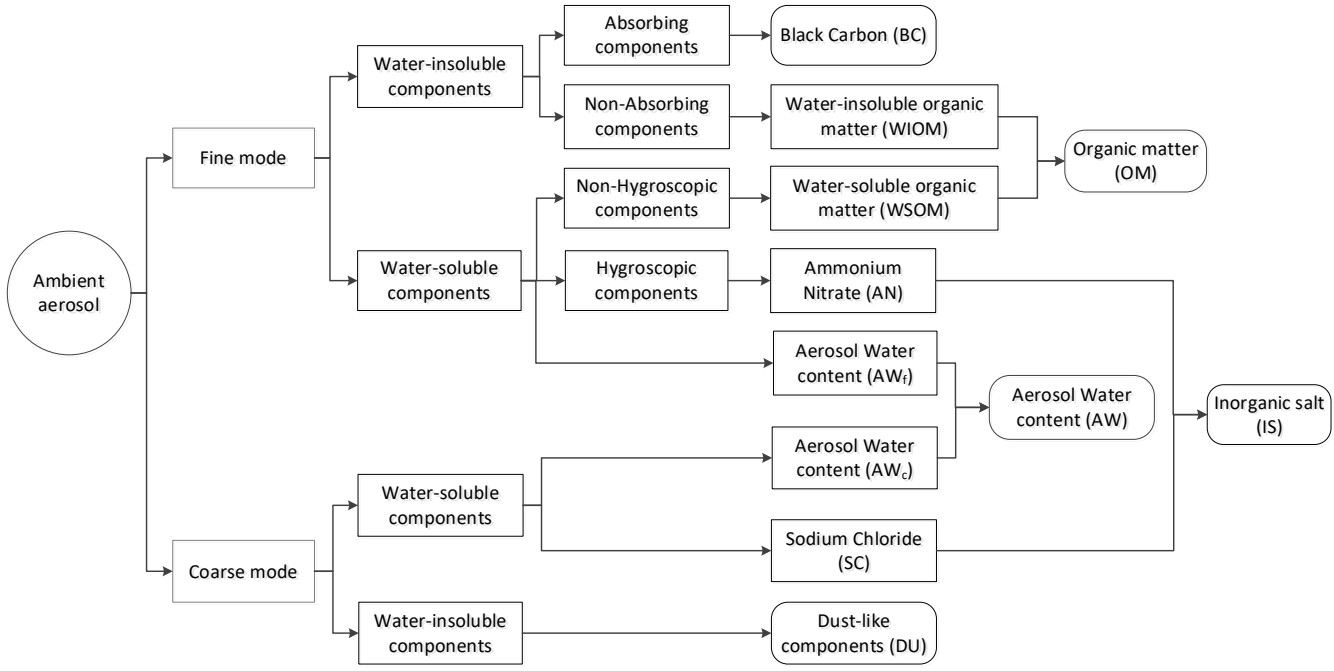


**Figure 3**. Aerosol component classification scheme.

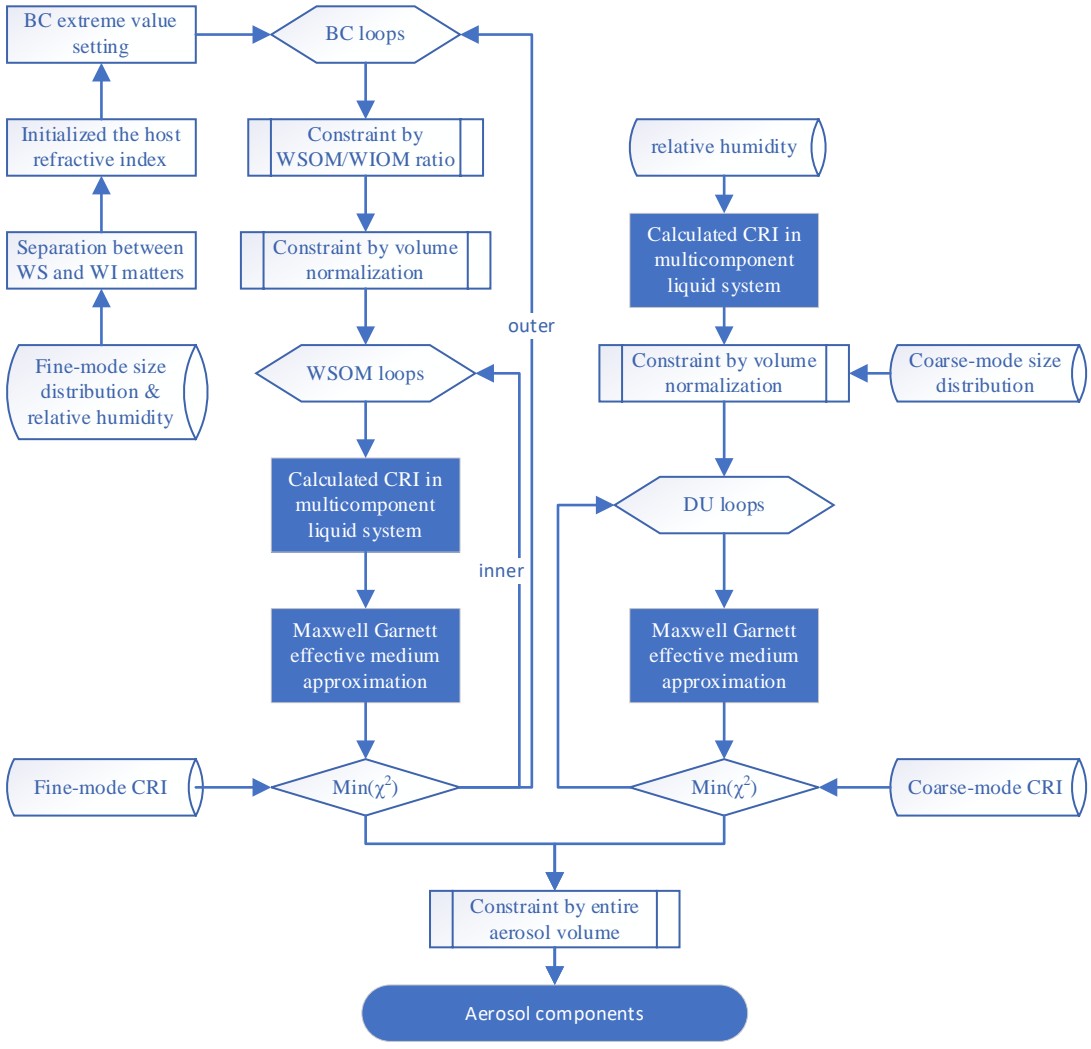

**Figure 4**. Flowchart of the aerosol component classification inversion algorithm.


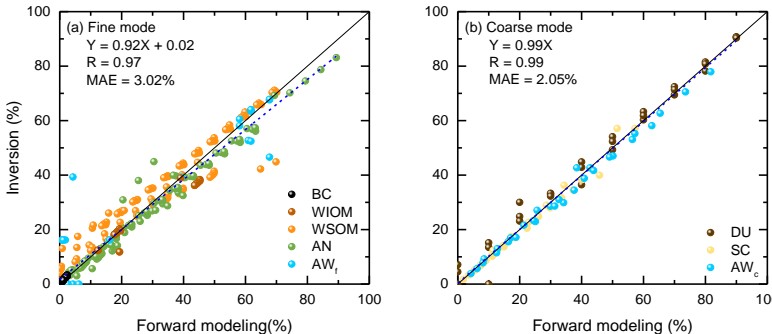

**Figure 5**. Scatter plots of volume fractions of aerosol components in the fine (left) and coarse (right) modes retrieved using the algorithm described in Chapter 3, versus those used as input calculated with the forward model. The solid line is the 1:1 line, and the dash line is the fitting line.

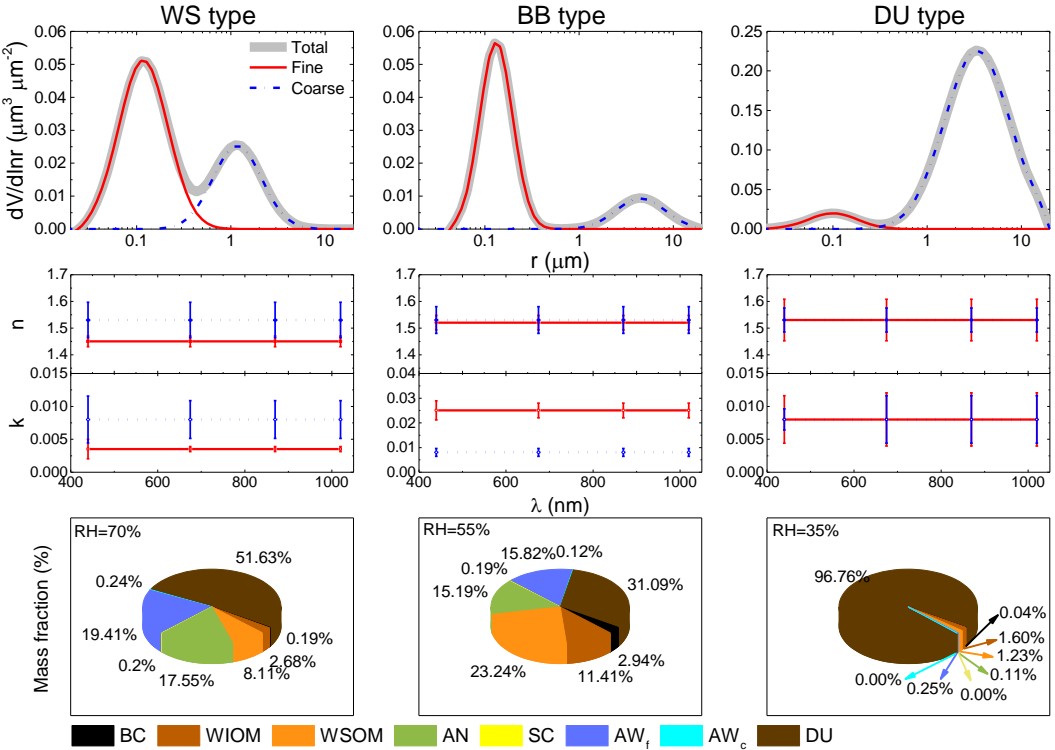

**Figure 6**. The fine and coarse-mode volume size distribution, complex refractive index and aerosol components describing the aerosol models used in the synthetic case study (WS: water soluble, BB: biomass burning, DU: dust).

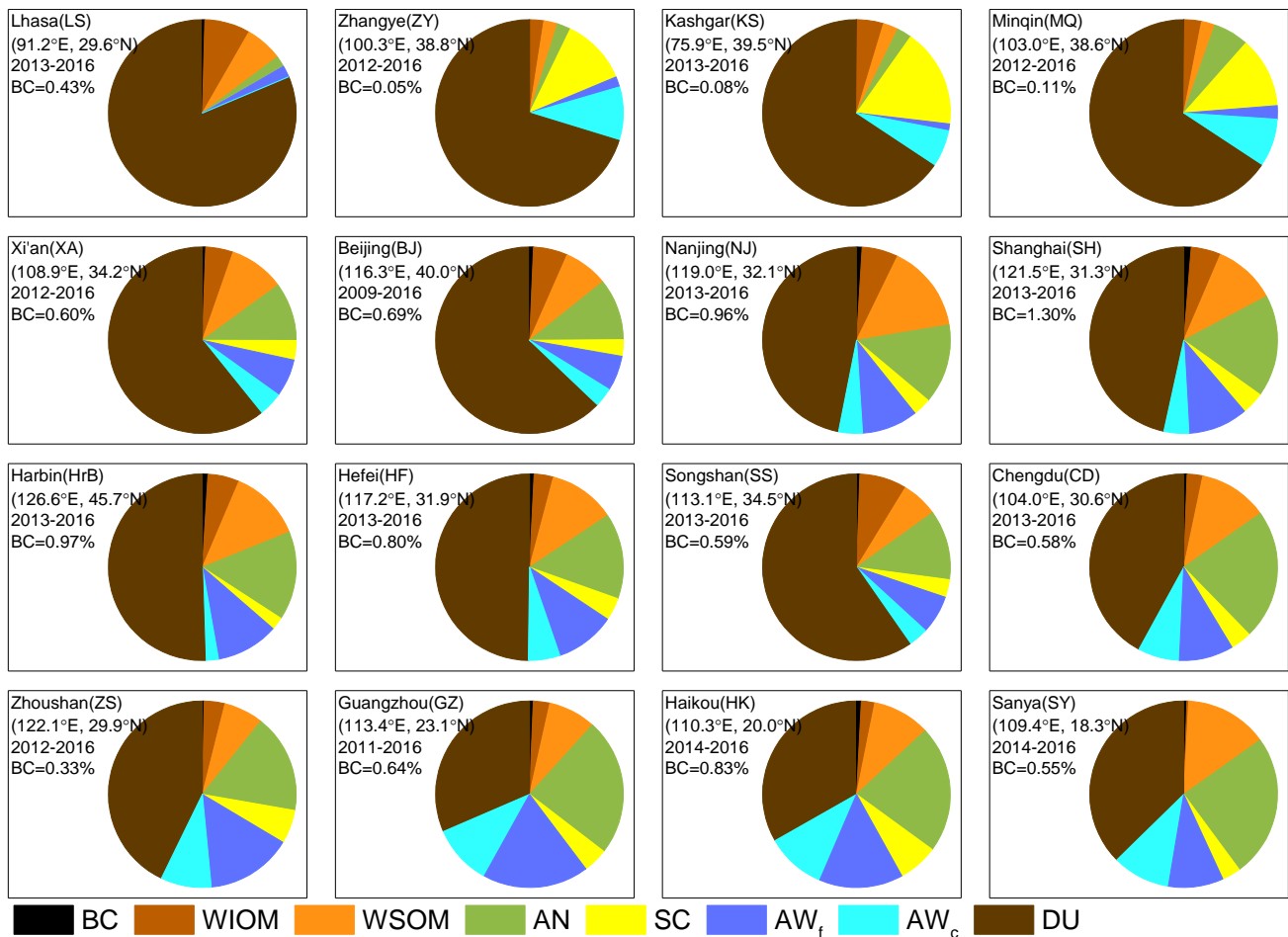


**Figure 7**. The averaged mass fraction of aerosol components at SONET sites. The site name, location, observation period and BC fraction are marked in each subgraph. The mass fractions of other components are listed in table S3.


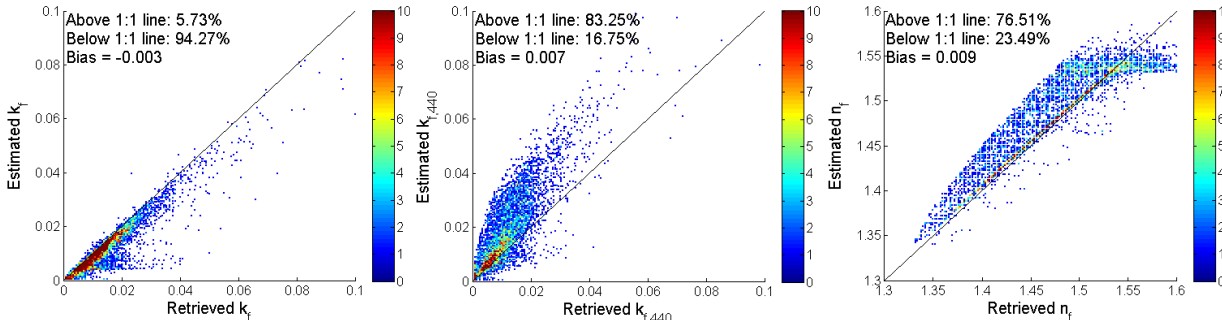

**Figure 8**. Data pair frequency of instantaneous imaginary parts of the complex refractive index at 675 nm ($k_f$), 440 nm ($k_{f,440}$), and real part at 440 nm ($n_f$) which are sorted according to ordered pairs (Retrieved, Estimated) in 0.0005 and 0.001 intervals for imaginary and real parts, respectively. "Retrieved" represents sub-component of CRI from the optical-physical retrievals, and "Estimated" is estimated by retrieved chemical components. The color represents the number of cases (color bar), and the solid black line shows the 1:1 line.

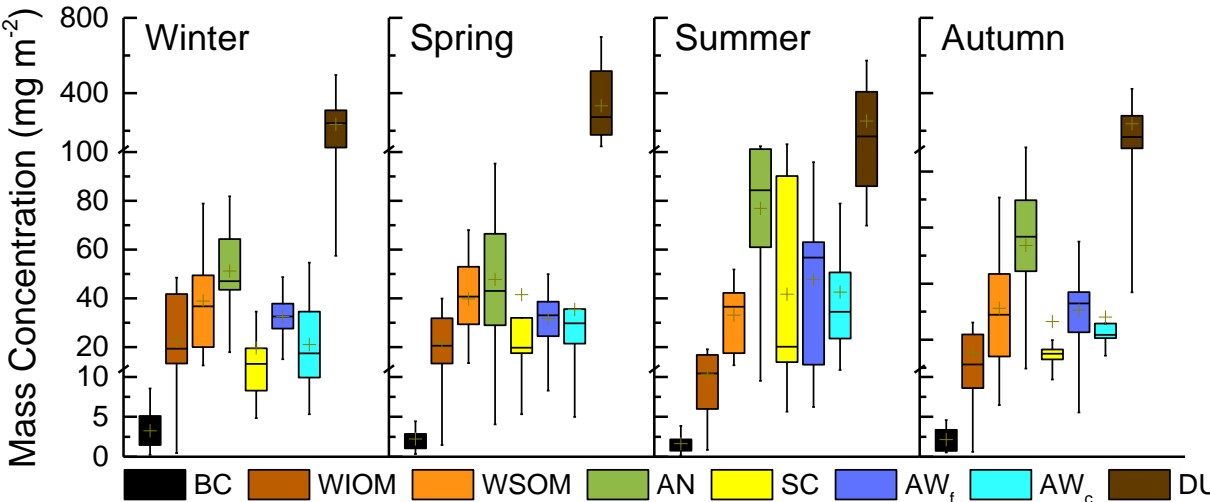

**Figure 9**. The mass concentrations of aerosol components in four seasons (winter, spring, summer and autumn). For the box-whisker plot, the mean value is indicated by a plus sign (+), and the median value by a short line inside the box (−). The top and bottom edges of each box represent the top and bottom quartiles (Q3 and Q1), and the corresponding whiskers are the outliers (Q3+1.5IQR and Q1-1.5IQR, IQR is interquartile range).

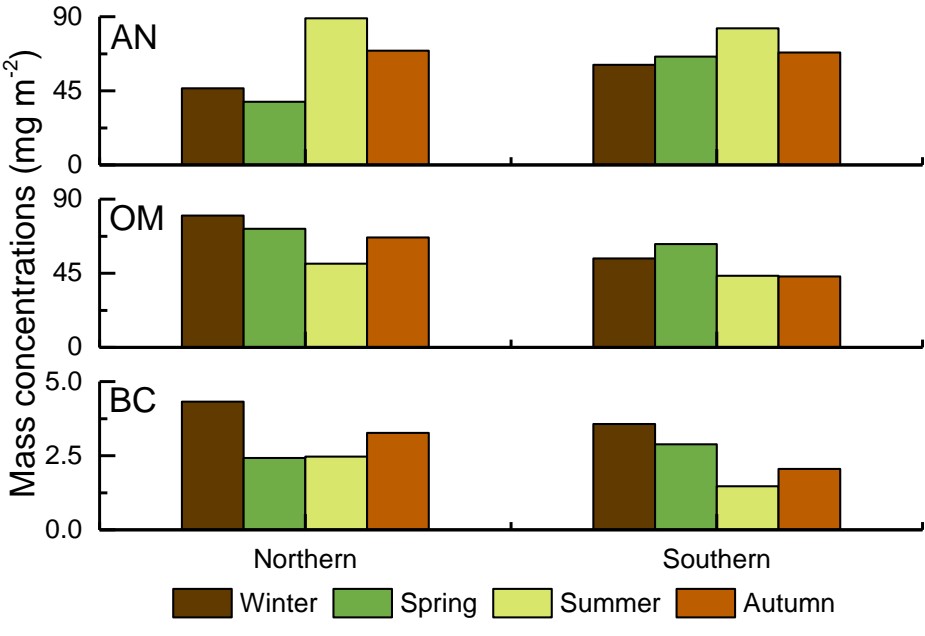


Figure 10. Comparison of aerosol component mass concentrations in northern (Xi'an, Beijing, Harbin, Hefei and Songshan) and southern China (Nanjing, Shanghai, Zhoushan, Guangzhou, Haikou and Sanya).

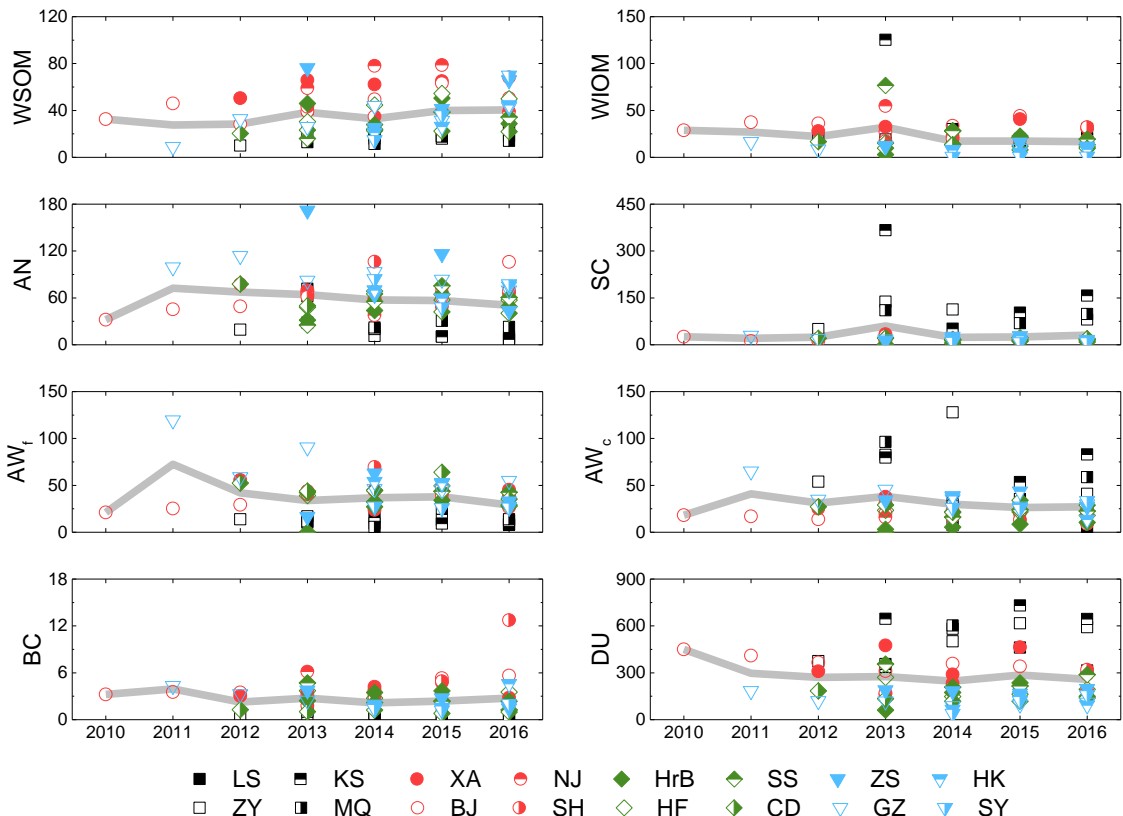

**Figure 11**. The interannual variations of mean aerosol component mass concentrations integrated over the whole atmospheric column with SONET sites from 2010-2016. The gray line represents the mean mass concentration of aerosol components averaged over the 16 sites; the points in each graph show the yearly value at each site. The abbreviations for the site names are from table 1.