# Peer review of "Improved inversion of aerosol components in the atmospheric column from remote sensing data"

_Atmospheric Chemistry and Physics, 2019_

## Referee Comment (RC1) · Anonymous Referee #1 · 19 Feb 2020

This paper expands upon the authors' previous aerosol-components retrieval (Zhang et al., Atm Env, 2018) by including sodium chloride as a coarse aerosol component. The authors apply their results to about 16 SONET sites all across China. The grammar is clear and for the most part the paper is very well written. This is a good paper that is suitable for publication in ACP after some modifications.

The authors cite Zhang 2018 for their methodology, but I am not exactly sure of their approach. I gather that they use the Zhang 2018 approach to determine separate complex refractive indices (CRI) for the fine and coarse modes from the SONET data. Then for the coarse mode, they use RH to determine the equilibrium mixture ratio of NaCL with water, which has a certain real refractive index (RRI). Once the RRI for the water-NaCl mixture is known, they can iterate the dust mixing ratio until they minimize the $\chi^2$

of Eq 12. They are using a single "dust," though, so they can not vary the IRI independent of the RRI; thus, they have limited adjustability for the spectral dependence of the CRI. This is all very reasonable, but the use of a single "dust" will sometimes increase their residuals. That is ok, though, as residual values can be monitored and retrievals can be rejected on the basis of residual values when necessary. I am comfortable with their coarse mode methodology.

I am having difficulty understanding the fine mode retrieval methodology, though, which is my biggest reservation about this paper. The authors claim to separate water-soluble organic carbon (WSOC) from ammonium nitrate (AN), but it is not clear to me how this can be accomplished without a specific assumption for the hygroscopicity parameter (kappa) of WSOC. If this is what the authors are doing, they need to specifically state this and provide the reader with the value of kappa that they chose for WSOC (as well as the rationale for using a certain kappa, and some discussion of the repercussions of using the wrong kappa in their retrieval). The authors cite (Zhang 2018), but a brief overview of the Zhang approach for the fine mode in the methodology section would be helpful.

*Major Issues*

It is not clear to me how the "derived hygroscopic parameter kappa" is obtained (p2, line 59, and Table 2). I believe the authors are deriving the Table 2 values from Equation 4, but that requires the hygroscopicity parameters of the components ($\kappa_i$); the authors say that these values can be computed by the component hygroscopic growth factors (lines 144-145 and Eq 5). However, I don't see how these component growth factors can be derived from their data, so I am assuming that they are obtaining these values from the literature. If this is the case, the authors should provide the reader with the $GF_i$ or $\kappa_i$ that they use in the retrieval. Otherwise, they should provide additional details about how they obtained the $\kappa_i$ with the sunphotometer data.

Figure 3:

I was confused by the "non-hygroscopic components" that are a subset of the "water-soluble components" and are the entire basis of the "water-soluble organic matter (WSOM)" — I am not a chemist, so I found it odd that water-soluble aerosols could be non-hygroscopic. It would be helpful to some readers (like myself) if the authors spent a sentence or two alerting the reader that water-soluble aerosols are sometimes non-hygroscopic. If they can explain the physics behind this phenomena, that would be even better.

Personally, I am skeptical about separating WSOM from AN using remote sensing techniques. From an optical standpoint, such a mixture would merely be a solution with an effective hygroscopicity parameter (kappa). Knowledge of RH and an assumed kappa allows one to derive the solute mixing ratio (and growth factor) via Eq 2, but I don't see how one can separate the effects of multiple soluble components with the available remote sensing information (refractive index and RH) without additional assumptions (like the $\kappa_i$ for each component).

Line 115, authors state:
**"For fine mode, the water-insoluble and water-soluble components are identified using an empirical function (Zhang et al., 2018)"**

How? The authors need to expand this a little. I checked the Zhang 2018, and I was not able to quickly determine how WI and WS components were separated. At a minimum, the authors should point to the specific section number in Zhang (2018), but it would be best to provide the readers with a brief recapitulation in order to best hold their interest.

Section 3.3:
The forward model is described well in Section 3.2, but the inversion section (3.3) is very light. For instance, the authors cover the relationship of the real refractive index to molar refractivity in Sect 3.2, but none of that shows up in Section 3.3. Presumably the authors are using RH to partition between the soluble components and water and also to assign a RRI to the host solution prior to the minimization procedure described in

section 3.2 (which requires refractive indices of both the host solution and the insoluble inclusions). None of that is stated here, though, so as a reader I am not sure if I have this correct.

Table 3:
Why is the RRI of NaCl and the coarse water so uncertain? I thought we had some good measurements of these species. Even if we didn't, how do we get 900

*Minor Issues*

Page 1, line 28, authors state:
**"Optical remote sensing techniques do not provide sufficient information for a detailed analysis of chemical composition and therefore refrain to the retrieval of components describing specific properties"**

My interpretation of this sentence is that we can not retrieve aerosol composition from remote sensing techniques, but I am sure that is not the authors intent (otherwise, we don't need to read the paper). Consider rephrasing.

Page 2, line 37:
Schuster (2009) is entitled "Remote Sensing of Aerosol Water Uptake," and does not directly address dust.

Page 3, line 82:
Should also reference Dubovik, O., and M. King (2000), A flexible inversion algorithm for retrieval of aerosol optical properties from sun and sky radiance measurements, J. Geophys. Res., 105(D16), 20,673–20,696.

Page 3, lines 87-88.
I don't understand the meaning of these lines:
**"Using these data, PVSD and CRI sub-modal parameters of atmospheric aerosols are obtained using the modal decomposition method proposed by**

**Zhang et al. (2017). Using the sub-modal characteristics data set thus obtained, an aerosol sub-modal model was established for China by Li et al. (2019), but the submodal aerosol components have not been given."**

So a sub-model model was established but not given?

Page 6, line 150:
Please replace "refractive index" with "real refractive index." Page 6, line 156:
Please tell the reader that "n" is the real refractive index.

Equation 10:
Equation 4 uses the symbol epsilon as the component dry volume fraction, whereas here it is the permittivity. Need to change the symbol used for dry volume fraction in Eq 4 and everywhere.

Equation 12:
The numerator should be squared. Otherwise, large negative differences will produce the "best" $\chi^2$.

Table S1:
What is the basis for the numbers in Table S1? That is, which climatology are you using to define WS, BB, and DU?

Page 9, lines 238-241:
Does it make sense to quantitatively discuss BC in the context of Fig 5? BC barely exists in that figure. I recommend adding a table or an additional figure for BC.

Figure 6:
How is the color scale in Fig 6 normalized (range is 0 to 20)? Also, it is odd that some of the "estimated" values are so far off when you are using CRI as a constraint. The $\chi^2$ must be very high in these cases. It would make sense to have a residual requirement (i.e., $\chi^2 <$ some threshold) and to throw out high values of $\chi^2$. This should also improve your statistics (slope, intercept, bias, etc.).

Lines 248-249: Authors state
**"As shown in table 3, although the TRE of BC is the lowest, it also causes the largest $k_f$ and $k_{f,440}$ errors."**

TRE is total relative error, so how can TRE cause $k_f$ and $k_{f,440}$ errors? Shouldn't cause and effect be the other way around (i.e., k errors cause TRE errors).

Lines 252-253, Authors state:
**"This indicates that most inversion results have good optical closure, and the aerosol components retrieved by the remote sensing method used in this study should be reasonable."**

This line refers to Fig 6, which is a plot of how well the component-averaged imaginary index compares to the imaginary refractive index that is used as input. Thus, a good comparison just means that you usually have good residuals (i.e., low $\chi^2$). Fig 6 does not assure reasonableness of all components in the retrieval, though, as it only shows the imaginary RI, and most of the components of this retrieval are not sensitive to IRI. The only thing that we can claim via Fig 6 is that the retrieval might be getting BC correct. Additionally, we can't use Fig 6 to argue that the BC mass or volume fractions are correct, as these are sensitive BC refractive index. However, you can use Fig 6 to argue that you are getting the BC AAOD correct; this is because you are using IRI as a constraint, and the IRI that you retrieve will always be the same (as long as BC has a spectrally flat IRI and your other absorbers do not).

Line 275:
I believe that the word "autumn" should be replaced with "spring."

Table 3:
Total relative error is defined with 7 parameters. Presumably this is $n_f$, $k_f$, $k_{f,440}$, $n_c$, $k_c$, $k_{c,440}$, and RH. Are the $(n_f, k_f)$ averaged from the 675, 870, and 1020 nm wavelengths, then? I don't recall seeing this explicitly stated.

Figure 2:

Caption should describe the timeframe of the boxplots.

Figure 4:

Do the pie charts correspond to both the fine and coarse modes? If so, why isn't there any AWc or SC in the WS and BB pie charts? If not, why does dust dominate over WIOM for those species?

Figure 7:

Throughout the text, authors use SC for sodium chloride. Here, they do not show SC but show SS (sea salt?).
* * *

---

## Referee Comment (RC2) · Anonymous Referee #3 · 4 May 2020

General comments:

Article is well in the scope of the ACP journal and discusses a method for retrieval of aerosol chemical composition from remote sensing. Aerosol chemical composition is in a grand demand by a scientific society providing data to evaluate global atmospheric modelling and climatic models. Presented method introduced some of the novelties and meanwhile is based on a well-established and proven techniques. Paper is well referenced at least to my knowledge, although I've found some inaccuracies (see in notes below).

At the same time, I have a feeling that paper doesn't reflect fully the potential of the method and the complexity of the work performed by authors, and definitely doesn't provide enough explication to reproduce the described method. Article is very brief and

[Figure]

dry, to the extent it sometimes hard to read and what is more important to understand concepts of the study preformed. Also, I've found out quite a number (for a short paper of 10 pages) of typos, please, consider general grammar revision, some of them I've listed below.

I would recommend this article for publication after major revision, which in my opinion should increase the impact and significance of the presented results.

Below I've highlighted some points where there is a room for improvement.

Major comments:

Formula 12: Possible typo, square is missing in numerator. Either it is missing either it is no chi-squared function. Please, clarify. If no square formula was actually used for the study, I'm not sure the method is legit as opposite sign errors in refractive indices at different wavelengths can compensate each other. Minimisation procedure of chi2 function is not described at all, at least mention how it is done. What is "iterative kernel function", Is it LUT or a version of Newtonian method? is it the same that was used in previous studies? Please clarify or refer.

Whole section 3 is rather confusing, despite of illustrations and formulas it doesn't give a clear understanding of a methodology, just some of its pieces. Also, if this method is an improvement of an existing one, it should be clearly referred and changes introduced highlighted. Please, consider re-formulating this part to make it clearer.

Figure 3 and Aerosol classification in section 3.1, it is hard to understand at which part of the retrieval this classification is used. And what parameters it used to classify aerosol in groups? Is it part of forward modelling? Does it use PVD provided by SONET, other part of paper claims only refractive index is used... And section 3.3 refers to refractive index of fine and coarse mode...

Also "combined" aerosol types such as BC, OM, IS and AW are used only in figure 3 and not referenced anywhere else in the paper. Why then there are presented?

Personally, I would prefer here a flowchart of the retrieval algorithm in general, i.e. where it would be easy to understand what parameters enter it, how refractive index is modelled and fitted and what parameters are coming out.

Error analysis: I would like to be convinced that method works, by showing a retrieval without any noise added, proving that it retrieves the exact pre-defined composition, i.e. calculating CRI using the forward model and retrieving it back again. Errors are analysed only from the point of view of data uncertainties, although some of them can emerge from the retrieval itself (i.e. inaccuracy of the forward model) and an obvious fitting bias in figure 6 can be an indication of that.

Data analysis: Again, very dry, some additional analysis (for e.g. splitting sites into several groups and analysing the seasonal averages for the groups) would be appreciated. For instance, there was a statement that higher BC in winter is because of heating in north region, splitting sites into heated/not heated regions and analysing seasonal trend would make this statement much more trustworthy and results obtained more significant. Also, a comparison to a previous method could be presented, if, of course, such comparison could be done. For e.g. comparison with OM from Zhang 2018 could be performed to illustrate improvements (if any), or discuss the similarities/differences observed.

There is a misfit error available as a result of the retrieval, why not use it to clean up the data a bit? This will provide more trustworthy results. Besides, as I understood estimations of the SONET retrieval errors for complex refractive index are available, why not get rid of the results whose fits are below the error bars? Authors themselves claim that ∼40% of the retrieval are above the average error, i.e. these results have questionable quality and could significantly influence the statistics provided.

Minor comments:

Line 70: I would suggest replacing "much information" to "sufficient information"

[Figure]

Lines 87-89: "Using the sub-modal characteristics data set thus obtained, an aerosol sub-modal model was established for China by Li et al. (2019), but the sub- modal aerosol components have not been given." I'm not sure what authors mean by this sentence, please, consider rephrasing.

Line 94: "…the linear interpolation method is used to match the SONET observations…", please clarify if interpolation in time or alt/lat/lon or all together were used.

Line 210. It is claimed that error of 50% in AWf is "acceptable". No references or desired thresholds were provided to jump to such conclusion. Consider providing them, or rephrasing the sentence to a milder comparison.

Line 308: "…improved a component inversion algorithm…", please refer to a baseline method. And what improvements authors refer to? No comparison to previous methods shown, is it complexity of the composition? Please clarify.

Line 353: I presume that another Dubovik 2000 paper, "Dubovik, O., and M. D. King, 2000: A flexible inversion algorithm for retrieval of aerosol optical properties from sun and sky radiance measurements. J. Geophys. Res., 105, 20 673–20 696." will be more suitable in the context it is referenced in the article.

Figure 5. I would appreciate to have numbers on pie charts too like figure 4, or an additional table with percentage of the components for each site.

Figure 6. Why only imaginary for fine and only 670? Fitting statistics is a necessary (but not sufficient) metric to justify that the method works. Please consider showing at least a real part at 670 too, but I would be more convinced to see all of them, all wl and fine/coarse. At least show minimal set, following the structure of Table s2 fine/coarse real/imaginary and 440 too, since they all referenced to have different level of uncertainty. Also, please, mention that it is fine mode in figure caption.

Figure 7. Is it possible to re-plot or at least re-paint so the color scheme will be the

same as in figs 5 and 6?

Table 2. Please specify what "No" in the head of the table means. If this is a number of observations, then why for quite different observation periods the numbers are so close? Was observation data filtered? Please, clarify.

Table s2 and s1. They are not a part of a publication provided (.pdf) and I don't see much reason why. I mean they are rather small and multiply referenced from the text, what is the reason to have one external half-page containing them?

Technical comments:

Figure 3. Size distribution axis. I believe it is $\mu$m3*$\mu$m-2

Line 27-28: ". . .retrieval of the chemical composition. . ."

Line 49: ". . .simultaneously retrieved. . ."

Line 56: ". . .to solve for the refractive index in a multicomponent liquid system"

Line 57: ". . .in the algorithm. . ."

Line 70: ". . .radiation to determine the columnar water vapor"

Line 93-94: Probably author meant "To avoid observation uncertainties, only data from manned weather stations that are maintained regularly were used."

Line 185: ". . . by the different aerosol size distribution. . ."

Line 246: "On the one hand"

Line 247: " . . . than for the other components. . ."

Line 252: ". . . data points are below. . ."

Line 260: " . . . high in spring. . ."

Line 260: ". . . to the dust transport from the northwest China. . ."
Line 261: "...seasons indicates that the aerosol at some sites"

Line 262: " ...due to the observational errors..."

Line 304: "...changes in the meteorological conditions..."

———————————————————————

---

## Author Comment (AC1) · 15 Jun 2020

This paper expands upon the authors' previous aerosol-components retrieval (Zhang et al., Atm Env, 2018) by including sodium chloride as a coarse aerosol component. The authors apply their results to about 16 SONET sites all across China. The grammar is clear and for the most part the paper is very well written. This is a good paper that is suitable for publication in ACP after some modifications.

The authors cite Zhang 2018 for their methodology, but I am not exactly sure of their approach. I gather that they use the Zhang 2018 approach to determine separate complex refractive indices (CRI) for the fine and coarse modes from the SONET data. Then for the coarse mode, they use RH to determine the equilibrium mixture ratio of NaCL with water, which has a certain real refractive index (RRI). Once the RRI for the water-NaCl mixture is known, they can iterate the dust mixing ratio until they minimize the $\chi^2$ of Eq 12. They are using a single "dust," though, so they cannot vary the IRI independent of the RRI; thus, they have limited adjustability for the spectral dependence of the CRI. This is all very reasonable, but the use of a single "dust" will sometimes increase their residuals. That is ok, though, as residual values can be monitored and retrievals can be rejected on the basis of residual values when necessary. I am comfortable with their coarse mode methodology.

I am having difficulty understanding the fine mode retrieval methodology, though, which is my biggest reservation about this paper. The authors claim to separate water-soluble organic carbon (WSOC) from ammonium nitrate (AN), but it is not clear to me how this can be accomplished without a specific assumption for the hygroscopicity parameter (kappa) of WSOC. If this is what the authors are doing, they need to specifically state this and provide the reader with the value of kappa that they chose for WSOC (as well as the rationale for using a certain kappa, and some discussion of the repercussions of using the wrong kappa in their retrieval). The authors cite (Zhang 2018), but a brief overview of the Zhang approach for the fine mode in the methodology section would be helpful.

**Major Issues**

1.  It is not clear to me how the "derived hygroscopic parameter kappa" is obtained (p2, line 59, and Table 2). I believe the authors are deriving the Table 2 values from Equation 4, but that requires the hygroscopicity parameters of the components ($\kappa_i$); the authors say that these values can be computed by the component hygroscopic growth factors (lines 144-145 and Eq 5). However, I don't see how these component growth factors can be derived from their data, so I am assuming that they are obtaining these values from the literature. If this is the case, the authors should provide the reader with the GFi or $\kappa_i$ that they use in the retrieval. Otherwise, they should provide additional details about how they obtained the $\kappa_i$ with the sun photometer data.

**Response:** Thanks for the reviewer's comment. The values of $\kappa_i$ are obtained from the literature and listed in table 2. For clarity, we have revised the manuscript as follows:

Line 156-157: "…where $\kappa_i$ is the hygroscopic parameter of the $i$th component obtained from the literature (table 2), and $f_{dry, i}$ is the dry component volume fraction defined as … "

2.  Figure 3:

I was confused by the "non-hygroscopic components" that are a subset of the "water-soluble components" and are the entire basis of the "water-soluble organic matter (WSOM)" ˘A˘T I am not a chemist, so I found it odd that

water-soluble aerosols could be non-hygroscopic. It would be helpful to some readers (like myself) if the authors spent a sentence or two alerting the reader that water-soluble aerosols are sometimes non-hygroscopic. If they can explain the physics behind this phenomena, that would be even better.

Personally, I am skeptical about separating WSOM from AN using remote sensing techniques. From an optical standpoint, such a mixture would merely be a solution with an effective hygroscopicity parameter (kappa). Knowledge of RH and an assumed kappa allows one to derive the solute mixing ratio (and growth factor) via Eq 2, but I don't see how one can separate the effects of multiple soluble components with the available remote sensing information (refractive index and RH) without additional assumptions (like the _i for each component).

**Response:** For the non-hygroscopicity of organic components, like vinegar and alcohol, they are water-soluble but more volatile than hygroscopic. In aerosols, the dicarboxylic acids are dominant in the WSOM component, and oxalic acid accounts for more than 50% in dicarboxylic acids, followed by succinic, malonic, maleic, adipic and phthalic acids (Chebbi & Carlier, 1996). Under ambient relative humidity (RH) of 10 - 90%, oxalic acid hardly deliquesce and exhibit low hygroscopicity, and others also have low hygroscopicity except malonic acid as shown in Figure R1 (Ma et al., 2013; Drozd et al., 2014; Jing et al., 2016). Considering the abundance of oxalic acid (more than 50% in dicarboxylic acids) and other low hygroscopic components in the atmospheric aerosols (Sullivan et al., 2009; Fu et al., 2013), WSOM is treated as a non-hygroscopic component in aerosol particles. We explained this point in the manuscript as follows:

Line 124-128: "It should be noted that the water-soluble property of aerosol components is not equivalent to hygroscopicity. Dicarboxylic acids represented by oxalic acid are dominant in the WSOM component but their hygroscopicity is extremely low (Ma et al., 2013; Drozd et al., 2014; Jing et al., 2016). Also other organic compounds in aerosols are less hygroscopic as shown in Zhang et al. (2018) (their figure 1). Hence, the OM components (WSOM and WIOM) are treated as non-hygroscopic components."

[Figure]

Figure R1. (a) Hygroscopic growth factors of oxalic acid from figure 2 of Jing et al. (2016); (b) Different geometric hygroscopic growth factor between inorganic and organic aerosols with RH around 85% of relative humidity from figure 1 of Zhang et al. (2018).

For the separation of the WSOM and AN components, the refractive index (RI) from remote sensing is used indirectly. From Eq. 2-5, we can get the relationship among the volume fraction of the components in the host. Because the sum of the volume fractions of these components can be equal to 1 in the host, the RI of the host can be calculated using Eq. 6-9. And then we can calculate the RI of the entire particle using the effective medium approximation (Eq.

10-12). Adjusting this fraction and comparing with the RI from the SONET observations, the volume fraction for all components can be obtained. The description and flowchart of the inversion procedure are added in section 3.4.

Ma Q., He H., Liu C. (2013). Hygroscopic properties of oxalic acid and atmospherically relevant oxalates, Atmospheric Environment, 69, 281-288.

Drozd G., Woo J., Hakkinen S.A.K., Nenes A., and McNeill V.F. (2014). Inorganic salts interact with oxalic acid in submicron particles to form material with low hygroscopicity and volatility, Atmos. Chem. Phys., 14, 5205–5215.

Chebbi A. & Carlier P. (1996). Carboxylic acids the troposphere, occurrences, sources, and sinks: a review, Atmospheric Environment, 30(24), 4233-4249.

Jing B., Tong S., Liu Q., Li K., Wang W., Zhang Y. and Ge M. (2016). Hygroscopic behavior of multicomponent organic aerosols and their internal mixtures with ammonium sulfate, Atmos. Chem. Phys., 16, 4101–4118.

Fu, P., Kawamura, K., Usukura, K., and Miura, K.: Dicarboxylic acids, ketocarboxylic acids and glyoxal in the marine aerosols collected during a round-the-world cruise, Mar. Chem., 148, 22–32, 2013.

Sullivan, R. C., Moore, M. J. K., Petters, M. D., Kreidenweis, S. M., Roberts, G. C., and Prather, K. A.: Timescale for hygroscopic conversion of calcite mineral particles through heterogeneous reaction with nitric acid, Phys. Chem. Chem. Phys., 11, 7826–7837, 2009.

3. Line 115, authors state:

**"For fine mode, the water-insoluble and water-soluble components are identified using an empirical function (Zhang et al., 2018)"**

How? The authors need to expand this a little. I checked the Zhang 2018, and I was not able to quickly determine how WI and WS components were separated. At a minimum, the authors should point to the specific section number in Zhang (2018), but it would be best to provide the readers with a brief recapitulation in order to best hold their interest.

**Response:** Thanks for the reviewer's comment. We added an explanation of the empirical function in the manuscript and add details to the supplementary as follows:

Line 115-118: "For the fine mode fraction, the water-insoluble and water-soluble components are identified using an empirical function (see section 2.2.2 in Zhang et al., 2018), which describes the ratio of the water-soluble to the water-insoluble volume fractions determined by RH, together with the parameterization of aerosol soluble volume fractions by Kandler and Schutz (2007)."

4. Section 3.3:

The forward model is described well in Section 3.2, but the inversion section (3.3) is very light. For instance, the authors cover the relationship of the real refractive index to molar refractivity in Sect 3.2, but none of that shows up in Section 3.3. Presumably the authors are using RH to partition between the soluble components and water and also to assign a RRI to the host solution prior to the minimization procedure described in section 3.2 (which requires refractive indices of both the host solution and the insoluble inclusions). None of that is stated here, though, so as a reader I am not sure if I have this correct.

**Response:** Thanks for the reviewer's comment. We adjusted the structure of the methods section and added a

description of the inversion procedure in both manuscript and supplementary as follows:

Line 179-216:

**"3.3 Effective medium approximation**

[revised manuscript text omitted]

5. Table 3:

Why is the RRI of NaCl and the coarse water so uncertain? I thought we had some good measurements of these species. Even if we didn't, how do we get 900

**Response:** Thanks for pointing this out. This large error (more than 900%) occurs only when the input error is large. For separation of high scattering components (NaCl and coarse water), the relative humidity (RH) and real part of CRI in coarse mode ($n_c$) are important dependent parameters and weakly correlated with other parameters. A large error for RH (10%) is applied to estimate the scattering components, which can be reduced because the actual observation error is usually lower than 5%. For the error of $n_c$ summarized from Zhang et al. (2017) listed in table S2, it is large for the WS typical model, which can also affect the scattering component separation. However, if the accuracy of RH is improved, the errors of NaCl and coarse water can be effectively suppressed. Meanwhile, it should be noted that this estimation of uncertainty is for a single measurement. One of important advantages of a remote sensing approach is to perform multiple measurements quickly in a short time period. Thus, the average uncertainty of the aerosol components can be effectively reduced taking account independent errors in each observation. We illustrate this point in the manuscript:

Line 270-275: "Fortunately, the RH observed by ground-based stations is accurate, with an error which is usually less than about 5% (WMO, 2008), which can significantly reduce the uncertainty in the retrieved aerosol scattering components. It should be noted that the uncertainties in table 3 are for single measurements. One important advantage of remote sensing is that multiple measurements can be made during a short period of time. Thus, the average uncertainty of the aerosol components can be effectively reduced by taking into account independent errors in each observation. In addition, the accuracy of the retrieved $n_c$ needs to be improved in order to deal with the aerosol component inversion."

**Minor Issues**

1.  Page 1, line 28, authors state:

**"Optical remote sensing techniques do not provide sufficient information for a detailed analysis of chemical composition and therefore refrain to the retrieval of components describing specific properties"**

My interpretation of this sentence is that we can not retrieve aerosol composition from remote sensing techniques, but I am sure that is not the authors intent (otherwise, we don't need to read the paper). Consider rephrasing.

**Response:** Thanks for reviewer's comment. This sentence was deleted.

2.  Page 2, line 37:

Schuster (2009) is entitled "Remote Sensing of Aerosol Water Uptake," and does not directly address dust.

**Response:** Thanks for reviewer's comment. We revised this sentence:

Line 36-41: "In a follow-up study, Schuster et al. (2009) applied a similar procedure to determine the aerosol water

fraction by fitting the real part of the refractive index of an internal mixture of water, soluble and insoluble species to observations by minimizing the cost function at all four wavelengths together. In this work the ratio of the dry volume fraction of insoluble to that of soluble aerosols was constrained by using a climatological value and the real refractive index which also prescribes the aerosol hygroscopicity. This constraint also provides a maximum insoluble fraction and the fraction of dust aerosol."

3.  Page 3, line 82:

Should also reference Dubovik, O., and M. King (2000), A flexible inversion algorithm for retrieval of aerosol optical properties from sun and sky radiance measurements, J. Geophys. Res., 105(D16), 20,673–20,696.

**Response:** We add this reference in manuscript.

Line 85-86: "Based on the inversion algorithm of Dubovik and King (2000) and Dubovik et al. (2000), the 440, 675, 870 and 1020 nm wavebands are used to … "

4.  Page 3, lines 87-88.

I don't understand the meaning of these lines:

"Using these data, PVSD and CRI sub-modal parameters of atmospheric aerosols are obtained using the modal decomposition method proposed by Zhang et al. (2017). Using the sub-modal characteristics data set thus obtained, an aerosol sub-modal model was established for China by Li et al. (2019), but the submodal aerosol components have not been given."

So a sub-model model was established but not given?

**Response:** We revised this sentence:

Line 91-93: "Using these fine and coarse mode characteristics of the CRI, micro-physical properties of aerosols in each mode were analyzed (Li et al., 2019), but the aerosol chemical components were not determined."

5.  Page 6, line 150:

Please replace "refractive index" with "real refractive index." Page 6, line 156:

Please tell the reader that "n" is the real refractive index.

**Response:** We revised as follows:

Line 162-164: "Firstly, the molar refractivity ($A_e$) at wavelength $\lambda$ can be calculated from the real part of the complex refractive index ($n_i$) and the volume fraction of the individual components"

6.  Equation 10:

Equation 4 uses the symbol epsilon as the component dry volume fraction, whereas here it is the permittivity. Need to change the symbol used for dry volume fraction in Eq 4 and everywhere.

**Response:** We change the symbol used for dry volume fraction to $f_{dry,i}$ in Eq 4 and 5. The corresponding descriptions have also been modified.

Line 153-160: "In the multicomponent liquid system, the hygroscopicity parameter $\kappa$ is given by the simple mixing rule

$$\kappa = \sum_i f_{dry,i}\kappa_i$$

(4)

where $\kappa_i$ is the hygroscopicity parameter of the $i$th component obtained from the literature (table 2), and $f_{dry,\,i}$ is the dry component volume fraction defined as

$$f_{dry,i} = \frac{V_i}{V_s}$$

"

7. Equation 12:

The numerator should be squared. Otherwise, large negative differences will produce the "best" $\chi^2$.

**Response:** Thanks for pointing this out. We missed the square when wrote the formula, but we used the formula where the numerator is the square.

8. Table S1:

What is the basis for the numbers in Table S1? That is, which climatology are you using to define WS, BB, and DU?

**Response:** Considering that the algorithm results from Dubovik et al. (2000) are taken as input in this study, the aerosol model is consistent with Dubovik's. In table S1, the parameters of size distribution are as same as those in table 2 of Dubovik et al. (2000), and the complex refractive index in the fine and coarse modes are further calculated using Dubovik's aerosol model from Zhang et al. (2017). We give more detail on this point in the supplementary.

9. Page 9, lines 238-241:

Does it make sense to quantitatively discuss BC in the context of Fig 5? BC barely exists in that figure. I recommend adding a table or an additional figure for BC.

**Response:** Thanks for reviewer's comment. All component fractions are listed in the table S3 and the BC fraction is added in figure 7 (Figure 5 from the previous manuscript) as follows:

[Figure]

**Figure 7**. The averaged mass fraction of aerosol components at SONET sites. The site name, location, observation period and BC fraction are marked in each subgraph. The mass fractions of other components are listed in table S3..

10. Figure 6:

How is the color scale in Fig 6 normalized (range is 0 to 20)? Also, it is odd that some of the "estimated" values are so far off when you are using CRI as a constraint. The $\chi^2$ must be very high in these cases. It would make sense to have a residual requirement (i.e., $\chi^2$ < some threshold) and to throw out high values of $\chi^2$. This should also improve your statistics (slope, intercept, bias, etc.).

**Response:** Thanks very much for your comments. We set a threshold and filter the results and revised as follows:

Line 303-321: "

The closure of the CRI between instantaneous optical-physical inversion and chemical estimation is examined by the data pair frequency. Figure 8 shows scatter density plots of the chemically estimated and sunphotometer-retrieved imaginary parts of the fine mode at 675 nm ($k_f$) and 440 nm ($k_{f,440}$) and the real parts of fine mode at 440 nm($n_f$). The points are colored by the number of data pairs (Retrieved, Estimated), which are sorted according to ordered pairs in 0.0005 intervals for the imaginary parts of CRI and 0.001 intervals for the real parts. The data pairs of $k_f$ are closely concentrated around the 1:1 line, although with slight underestimation with 94.3% of the estimated values lower than the retrieved values; only 5.3% of the data pairs have a relatively large absolute error (AE > 0.01). The mean bias is not large (-0.003), and the mean absolute value is equal to the mean absolute error (MAE = 0.003). There are two reasons for this slight underestimation in chemical estimation. On the one hand, the imaginary part

of the refractive index of BC is much larger than for the other components due to its strong absorption. Thus, the inversion of the BC concentration is very sensitive to the estimation of the refractive index. As shown in table 3, although the TRE of BC is the lowest, the errors caused by $k_f$ and $k_{f,440}$ are larger than for any other component. On the other hand, $k_f$ is not only affected by BC in the inversion process, but also affected by organic components (WSOM & WIOM) with spectral absorption characteristics. Therefore, in most cases, $k_f$ is underestimated in chemical estimation and $k_{f,440}$ is overestimated (Bias = 0.007). The mean relative error (RE) is 27.1%, and 62.8% of the data points are below the average relative error line. This indicates that most inversion results have good optical closure. For the closure of the real part of the fine mode, the data pairs of $n_f$ are also concentrated around the identity line, although 76.5% of the $n_f$ is above the identity line. Underestimation occurs mainly when $n_f$ is larger than 1.56, because the only component with the real part of the CRI larger than 1.56 is BC, but its concentration is mainly determined by the imaginary part. The bias of the estimated $n_f$ (Bias = 0.009) is larger than that of $k_f$ due to the fact that the value and the range of $n_f$ are larger than that of $k_f$. ”

[Figure]

**Figure 8**. Data pair frequency of instantaneous imaginary parts of the complex refractive index at 675 nm ($k_f$), 440 nm ($k_{f,440}$), and real part at 440 nm ($n_f$) which are sorted according to ordered pairs (Retrieved, Estimated) in 0.0005 and 0.001 intervals for imaginary and real parts, respectively. “Retrieved” represents sub-component of CRI from the optical-physical retrievals, and “Estimated” is estimated by retrieved chemical components. The color represents the number of cases (color bar), and the solid black line shows the 1:1 line.

11.  Lines 248-249: Authors state

**“As shown in table 3, although the TRE of BC is the lowest, it also causes the largest kf and kf;440 errors."**

TRE is total relative error, so how can TRE cause kf and kf;440 errors? Shouldn't cause and effect be the other way around (i.e., k errors cause TRE errors).

**Response:** We revised this sentence as follows:

Line 312-313: “As shown in table 3, although the TRE of BC is the lowest, the errors caused by $k_f$ and $k_{f,440}$ are larger than for any other component.”

12.  Lines 252-253, Authors state:

**“This indicates that most inversion results have good optical closure, and the aerosol components retrieved by the remote sensing method used in this study should be reasonable."**

This line refers to Fig 6, which is a plot of how well the component-averaged imaginary index compares to the imaginary refractive index that is used as input. Thus, a good comparison just means that you usually have good

residuals (i.e., low $\chi^2$). Fig 6 does not assure reasonableness of all components in the retrieval, though, as it only shows the imaginary RI, and most of the components of this retrieval are not sensitive to IRI. The only thing that we can claim via Fig 6 is that the retrieval might be getting BC correct. Additionally, we can't use Fig 6 to argue that the BC mass or volume fractions are correct, as these are sensitive BC refractive index. However, you can use Fig 6 to argue that you are getting the BC AAOD correct; this is because you are using IRI as a constraint, and the IRI that you retrieve will always be the same (as long as BC has a spectrally flat IRI and your other absorbers do not).

**Response:** Thanks for the reviewer's detailed explanation. We revised this sentence as follows:

Line 317: "This indicates that most inversion results have good optical closure."

13. Line 275:

I believe that the word "autumn" should be replaced with "spring."

**Response:** Thanks for pointing out this mistake. We revised as follows:

Line 342-343: "Low concentrations of BC in the other seasons are mainly due to the influence of frequent dust events in the spring and high aerosol hygroscopic growth in the summer."

14. Table 3:

Total relative error is defined with 7 parameters. Presumably this is nf, kf , kf;440, nc, kc, kc;440, and RH. Are the (nf ; kf ) averaged from the 675, 870, and 1020 nm wavelengths, then? I don't recall seeing this explicitly stated.

**Response:** Our method of separating fine and coarse modal CRI parameters (Zhang et al, 2017) can only separate 6 of these parameters (nf, kf , kf;440, nc, kc, kc;440). Thus we assume that the imaginary parts of CRI at other wavebands except 440nm are invariable. We add a statement for CRI sub-modal parameters as follows:

Lien 90-91: "The real parts of the CRI of the fine and coarse modes ($n_f$ and $n_c$, respectively) are spectrally independent, while the imaginary parts have spectral variation at 440 nm, so they are written as ($k_{f,440}$, $k_f$ ) & ($k_{c,440}$, $k_c$)."

15. Figure 2:

Caption should describe the timeframe of the boxplots.

**Response:** we revised the caption of figure 2 as follows:

 "**Figure 2**. Boxplots of the relative humidities observed near each of the SONET sites. The observation periods for each site are shown in table 1. The line and the diamond represent the median and mean values, respectively, and the box shows the standard deviation (1 σ)."

16. Figure 4:

Do the pie charts correspond to both the fine and coarse modes? If so, why isn't there any AWc or SC in the WS and BB pie charts? If not, why does dust dominate over WIOM for those species?

**Response:** Yes, this figure shows both the fine and coarse modes. Few studies have given the fraction of water content in coarse mode which SC is closely related to, so we set it very low in the typical model.

17.  Figure 7:

Throughout the text, authors use SC for sodium chloride. Here, they do not show SC but show SS (sea salt?).

**Response:** Thanks for pointing this out. We revised the figure.

---

## Author Comment (AC2) · 15 Jun 2020

Article is well in the scope of the ACP journal and discusses a method for retrieval of aerosol chemical composition from remote sensing. Aerosol chemical composition is in a grand demand by a scientific society providing data to evaluate global atmospheric modelling and climatic models. Presented method introduced some of the novelties and meanwhile is based on a well-established and proven techniques. Paper is well referenced at least to my knowledge, although I've found some inaccuracies (see in notes below).

At the same time, I have a feeling that paper doesn't reflect fully the potential of the method and the complexity of the work performed by authors, and definitely doesn't provide enough explication to reproduce the described method. Article is very brief and dry, to the extent it sometimes hard to read and what is more important to understand concepts of the study preformed. Also, I've found out quite a number (for a short paper of 10 pages) of typos, please, consider general grammar revision, some of them I've listed below.

I would recommend this article for publication after major revision, which in my opinion should increase the impact and significance of the presented results.

Below I've highlighted some points where there is a room for improvement.

**Major comments:**

Formula 12: Possible typo, square is missing in numerator. Either it is missing either it is no chi-squared function. Please, clarify. If no square formula was actually used for the study, I'm not sure the method is legit as opposite sign errors in refractive indices at different wavelengths can compensate each other. Minimisation procedure of chi2 function is not described at all, at least mention how it is done. What is "iterative kernel function", Is it LUT or a version of Newtonian method? is it the same that was used in previous studies? Please clarify or refer.

Whole section 3 is rather confusing, despite of illustrations and formulas it doesn't give a clear understanding of a methodology, just some of its pieces. Also, if this method is an improvement of an existing one, it should be clearly referred and changes introduced highlighted. Please, consider re-formulating this part to make it clearer. Figure 3 and Aerosol classification in section 3.1, it is hard to understand at which part of the retrieval this classification is used. And what parameters it used to classify aerosol in groups? Is it part of forward modelling? Does it use PVD provided by SONET, other part of paper claims only refractive index is used … And section 3.3 refers to refractive index of fine and coarse mode …

Also "combined" aerosol types such as BC, OM, IS and AW are used only in figure 3 and not referenced anywhere else in the paper. Why then there are presented?

Personally, I would prefer here a flowchart of the retrieval algorithm in general, i.e. where it would be easy to understand what parameters enter it, how refractive index is modelled and fitted and what parameters are coming out.

**Response:** Thank you very much for pointing this out.

(1) Formula 12 is indeed a typo. We have revised it in the manuscript as follows:

Line 211:"

$$\chi^2 = \sum_\lambda \frac{\left(m_{rtrl}(\lambda) - m(\lambda)\right)^2}{m_{rtrl}(\lambda)}$$

$\lambda$=440, 675, 870 and 1020 nm        (12)"

(2) In order to present our algorithm more clearly, we have added the flowchart in Figure 4 and corresponding descriptions (section 3.4).

Line 192-216:

**3.4 Inversion procedure**

The flow chart for the inversion of the aerosol components is shown in figure 4. In the fine mode, the ratio of WS and WI matter is estimated using RH as described in section 2.2.2 in Zhang et al. (2018). The initial value of the host refractive index and the extreme value for the BC component are set by the calculation modules of the complex refractive index in the multicomponent liquid system (see section 3.2) and the effective medium approximation (see section 3.3), respectively. In the loop to determine the BC component, two constraints are applied to separate BC from other components. The WSOM/WIOM ratio constraint was developed by Zhang et al. (2018) based on considerations published in the literature (Chalbot et al., 2016; Bougiatioti et al., 2013; Wozniak et al., 2013; Mayol-Bracero et al., 2002; Krivácsy et al., 2001; Zappoli et al., 1999):

$$\begin{cases} f_{WSOM} \cong \alpha f_{WIOM} \\ \alpha = \dfrac{\beta \rho_{WSOM}^{-1}}{1 - \beta \rho_{WSOM}^{-1}} \qquad \beta \in [44\%, 77\%] \end{cases}$$

(12)

For more detail, see section 2.3.1 in Zhang et al. (2018). The volume normalization of the aerosol components in both the fine and coarse modes is used to constrain the volume fraction of the aerosol components to a reasonable range (similar as section 2.3.2 in Zhang et al., 2018)

$$\begin{cases} f_{fine} + f_{coarse} = 1.0 \\ f_{fine} = f_{BC} + f_{AN} + f_{WSOM} + f_{WIOM} + f_{AW_f} \\ f_{coarse} = f_{DU} + f_{SC} + f_{AW_c} \end{cases}$$

(13)

Then the inner loop of WSOM computes the CRIs of the fine mode at different BCs, and output the aerosol components of minimum $\chi^2$. The inversion procedure for the coarse mode is simpler than that for the fine mode. There is only a loop for DU and the complex refractive index of the host can be directly calculated by equations (2) - (8) with only input of RH. The function Chi-squared ($\chi^2$) as an iterative kernel function is expressed in the sum of the differences between the complex refractive index estimated from the forward model ($m$) and the retrievals ($m_{rtrl}$), at multiple wavelengths:

$$\chi^2 = \sum_\lambda \frac{\left(m_{rtrl}(\lambda) - m(\lambda)\right)^2}{m_{rtrl}(\lambda)}$$

λ=440, 675, 870 and 1020 nm     (14)

The retrieval is completed when the value of $\chi^2$ reaches a minimum. The volume fractions of the aerosol components can be obtained by solving the above equations (10-12). The aerosol mass concentration in the atmospheric column is calculated using the volume and effective density of the aerosol components."

[Figure]

Figure 4. Flowchart of the aerosol component classification inversion algorithm .

(3) The "combined" aerosol types are used in table 3, and subsequent analysis (section 4) was associated with this combined component.

Error analysis: I would like to be convinced that method works, by showing a retrieval without any noise added, proving that it retrieves the exact pre-defined composition, i.e. calculating CRI using the forward model and retrieving it back again. Errors are analysed only from the point of view of data uncertainties, although some of them can emerge from the retrieval itself (i.e. inaccuracy of the forward model) and an obvious fitting bias in figure 6 can be an indication of that.

**Response:** Thank you for reviewer's comments. A set of retrievals without noise is added in the section 3.5 as follows: Line 226-240: "The uncertainty in the retrieval results was evaluated using synthetic data, both without and with input errors added. For the first case (without input errors), a set of complex refractive indices has been obtained by calculating a set of volume fractions of the aerosol components using the forward chemical model, which was used as input for the retrieval of the aerosol components without any noisy added. For the aerosol components, the volume fraction of BC was constrained between 0.0 to 3.0% with an interval of 0.5%, and corresponding dynamic ranges for the other components with intervals of 10%, in three ambient relative humidity conditions (40%,

60% and 80%). Figure 5 shows the comparison of the aerosol component volume fractions from forward modeling used as input, and their retrieved values. The volume fractions of the retrieved aerosol components are in good agreement with the input values. For the fine mode fraction, most data pairs are located close around the 1:1 line, with the mean absolute error (MAE) of the aerosol component volume fractions of 3.0%. In five samples the difference in the $AW_f$ is more than 20.0%, though the overall MAE for $AW_f$ is only 5.5%. In these five samples, the BC component is low and organic matter contributes substantially to the aerosol light absorption, resulting in underestimation of the $AW_f$ volume fraction at high RH and overestimation for moderate RH. WSOM is overall slightly overestimated and AN is underestimated by only a few percent. The correlation between the input and retrieved aerosol volume fractions in the coarse mode is even better than that in fine mode. The regression coefficient for all samples is 0.99, and the MAE is only 2.0%. These results show the very small uncertainty in the retrieved aerosol component volume fractions."

[Figure]

Figure 5. Scatter plots of volume fractions of aerosol components in the fine (left) and coarse (right) modes retrieved using the algorithm described in Chapter 3, versus those used as input calculated with the forward model. The solid line is the 1:1 line, and the dash line is the fitting line.

Data analysis: Again, very dry, some additional analysis (for e.g. splitting sites into several groups and analysing the seasonal averages for the groups) would be appreciated. For instance, there was a statement that higher BC in winter is because of heating in north region, splitting sites into heated/not heated regions and analysing seasonal trend would make this statement much more trustworthy and results obtained more significant. Also, a comparison to a previous method could be presented, if, of course, such comparison could be done. For e.g. comparison with OM from Zhang 2018 could be performed to illustrate improvements (if any), or discuss the similarities/differences observed.

**Response:** Thank you for the comment. We add the analysis about the seasonal variation of main aerosol compositions in fine mode between north and south China as follows:

Line 346-359: " The seasonal variation of the main aerosol components in the fine mode is discussed on a regional basis (Figure 10). BC concentrations in typical northern regions are higher than in southern regions, because of emissions due to winter heating only in the north. Other BC sources are vehicle emissions and biomass burning. Adverse meteorological conditions in winter result in the accumulation of BC in the atmosphere resulting in high BC values in both the north and the south. The highest BC mass concentrations in the northern region in the winter is 4.3 mg m$^{-2}$. OM is one of the dominant components in the fine mode, with sources similar to those of BC. The impact of biomass burning in the winter and spring over south China (Chen et al., 2017) is significant, leading to OM concentrations of more than 50.0 mg m$^{-2}$. In the northern region, much biomass burning occurs in the autumn (Wang et al., 2020). With the influence of heating, the OM level in the north can reach up to 80.1 mg m$^{-2}$. Therefore, the OM mass concentration in the northern region is only low in the summer (50.8 mg m$^{-2}$). AN is usually formed by secondary reactions of gaseous precursors in complex air

pollution areas. In both the northern and the southern region, AN mass concentration is larger in the summer than in other seasons, and the seasonal variation in the southern region is significantly smaller than that in the north. The mean AN mass concentration in the southern region is 8.7 mg m$^{-2}$ higher than that in the northern region. This suggests that more AN is produced by secondary reactions in the humid climate in the south than in the northern region."

[Figure]

Figure 10. Comparison of aerosol component mass concentrations in northern (Xi'an, Beijing, Harbin, Hefei and Songshan) and southern China (Nanjing, Shanghai, Zhoushan, Guangzhou, Haikou and Sanya).

Compared with Zhang et al. (2018), the main advantage of the new algorithm is the increased flexibility in selecting water-soluble components. The new algorithm gives a reasonable scheme of complex refractive index (CRI) estimation, considering the mixing of multiphase solution. We added an explanation in the revised manuscript as follows:

Line 216-224: "The retrieval algorithm described here is an improvement over that described in Zhang et al. (2018). In that algorithm, the WSOM component was added to the host, but it could only be considered as a non-hygroscopic component. The proportion of solute and solution in the host mixture at different relative humidities should be measured in the laboratory, which limits the choice of aerosol components in the inversion process. Also, the real part of the CRI of the host was calculated by volume averaging, which can introduce a small error. The improved algorithm described here is more suitable for the calculation of the properties of a mixture of multiple water-soluble components as long as the hygroscopic parameter is known, which is not only convenient to measure but also independent of particle size. The hygroscopic parameter of WSOM can be varied according to the choice of mixing components instead of changing the algorithm itself. Similarly, some other water-soluble components (e.g. sulfate) can be introduced into the inversion algorithm without laboratory measurements."

[Figure]

Figure R1. The differences of real part of CRI with relative humidity between the algorithm of Zhang et al (2018) and this study when the WSOM volume fraction is 10%, 50% and 80%.

There is a misfit error available as a result of the retrieval, why not use it to clean up the data a bit? This will provide more trustworthy results. Besides, as I understood estimations of the SONET retrieval errors for complex refractive index are available, why not get rid of the results whose fits are below the error bars? Authors themselves claim that ~40% of the retrieval are above the average error, i.e. these results have questionable quality and could significantly influence the statistics provided.

**Response:** Thanks very much for your comments. We set a threshold and filter the results and revised as follows:

Line 303-321: "The closure of the CRI between instantaneous optical-physical inversion and chemical estimation is examined by the data pair frequency. Figure 8 shows scatter density plots of the chemically estimated and sunphotometer-retrieved imaginary parts of the fine mode at 675 nm ($k_f$) and 440 nm ($k_{f,440}$) and the real parts of fine mode at 440 nm($n_f$). The points are colored by the number of data pairs (Retrieved, Estimated), which are sorted according to ordered pairs in 0.0005 intervals for the imaginary parts of CRI and 0.001 intervals for the real parts. The data pairs of $k_f$ are closely concentrated around the 1:1 line, although with slight underestimation with 94.3% of the estimated values lower than the retrieved values; only 5.3% of the data pairs have a relatively large absolute error (AE > 0.01). The mean bias is not large (-0.003), and the mean absolute value is equal to the mean absolute error (MAE = 0.003). There are two reasons for this slight underestimation in chemical estimation. On the one hand, the imaginary part of the refractive index of BC is much larger than for the other components due to its strong absorption. Thus, the inversion of the BC concentration is very sensitive to the estimation of the refractive index. As shown in table 3, although the TRE of BC is the lowest, the errors caused by $k_f$ and $k_{f,440}$ are larger than for any other component. On the other hand, $k_f$ is not only affected by BC in the inversion process, but also affected by organic components (WSOM & WIOM) with spectral absorption characteristics. Therefore, in most cases, $k_f$ is underestimated in chemical estimation and $k_{f,440}$ is overestimated (Bias = 0.007). The mean relative error (RE) is 27.1%, and 62.8% of the data points are below the average relative error line. This indicates that most inversion results have good optical closure. For the closure of the real part of the fine mode, the data pairs of $n_f$ are also concentrated around the identity line, although 76.5% of the $n_f$ is above the identity line. Underestimation occurs mainly when $n_f$ is larger than 1.56, because the only component with the real part of the CRI larger than 1.56 is BC, but its concentration is mainly determined by the imaginary part. The bias of the estimated $n_f$ (Bias = 0.009) is larger than that of $k_f$ due to the fact that the value and the range of $n_f$ are larger than that of $k_f$. "

[Figure]

**Figure 8**. Data pair frequency of instantaneous imaginary parts of the complex refractive index at 675 nm ($k_f$), 440 nm ($k_{f,440}$), and real part at 440 nm ($n_f$) which are sorted according to ordered pairs (Retrieved, Estimated) in 0.0005 and 0.001 intervals for imaginary and real parts, respectively. "Retrieved" represents sub-component of CRI from the optical-physical retrievals, and "Estimated" is estimated by retrieved chemical components. The color represents the number of cases (color bar), and the solid black line shows the 1:1 line.

In addition, I think what the reviewer said "Authors themselves claim that ~40% of the retrieval are above the average error" should refer to "Moreover, the mean relative error (RE) is 29.93%, and 61% of the data points is below the average relative error line." in the manuscript. It is not a constraint on the convergence of the algorithm, but to evaluate whether the convergence condition is reasonable. Therefore, it does not affect our statistical results.

**Minor comments:**

Line 70: I would suggest replacing "much information" to "sufficient information"

**Response:** We have revised in the manuscript as follows:

Line 73-74: "These radiation and polarization measurements can provide sufficient information to calculate the columnar aerosol optical depth (AOD) and further retrieve the aerosol microphysical parameters. "

Lines 87-89: "Using the sub-modal characteristics data set thus obtained, an aerosol sub-modal model was established for China by Li et al. (2019), but the sub-modal aerosol components have not been given." I'm not sure what authors mean by this sentence, please, consider rephrasing.

**Response:** We rephrase this sentence as follows:

Line 91-93: "Using these fine and coarse mode characteristics of the CRI, micro-physical properties of aerosols in each mode were analyzed (Li et al., 2019), but the aerosol chemical components were not determined."

Line 94: "… the linear interpolation method is used to match the SONET observations …", please clarify if interpolation in time or alt/lat/lon or all together were used.

**Response:** This linear interpolation is only used in timescale. We clarify this point in manuscript:

Line 99: " The CMA stations closest to each SONET site were selected and the meteorological data were collocated in time with the SONET observations by linear interpolation between the nearest observations."

Line 210. It is claimed that error of 50% in AWf is "acceptable". No references or desired thresholds were provided to jump to such conclusion. Consider providing them, or rephrasing the sentence to a milder comparison.

**Response:** The revised sentence is as follows:

Line 268: "Affected by SC, the TRE of $AW_c$ is also large due to $n_c$, but the TRE of $AW_f$ is much smaller (50.05%). "

Line 308: "… improved a component inversion algorithm …", please refer to a baseline method. And what improvements authors refer to? No comparison to previous methods shown, is it complexity of the composition? Please clarify.

**Response:** This study improves the algorithm of Zhang et al. (2018), not only for multiphase solution calculation, but also for adding SC component. We revised this sentence as follows:

Line 389-390: "In the current study, we updated the refractive index calculation in a multi-component liquid system and improved the component inversion algorithm of Zhang et al. (2018) to retrieve atmospheric columnar aerosol components including …"

Line 353: I presume that another Dubovik 2000 paper, "Dubovik, O., and M. D. King, 2000: A flexible inversion algorithm for retrieval of aerosol optical properties from sun and sky radiance measurements. J. Geophys. Res., 105, 20 673¨C20 696." will be more suitable in the context it is referenced in the article.

**Response:** Thank you for your comment. We add this reference in manuscript.

Figure 5. I would appreciate to have numbers on pie charts too like figure 4, or an additional table with percentage of the components for each site.

**Response:** Because of the small space in the figure, we only increased the volume ratio of BC in the blank space. The fractions of other components are listed in the table S3 of supplementary as follows:

**Table S3. The averaged mass fraction of aerosol components at SONET sites shown in figure 7.**

| Site | Fine mode | | | | | Coarse mode | | | AW | IS | OM |
|------|-----------|------|------|------|-------|-------------|------|--------|------|------|------|
| | BC | WIOM | WSOM | AN | $AW_f$ | DU | SC | $AW_c$ | | | |
| Lhasa | 0.43% | 7.85% | 6.47% | 1.89% | 2.01% | 81.34% | 0.00% | 0.00% | 2.01% | 1.89% | 14.33% |
| Zhangye | 0.05% | 2.43% | 2.36% | 2.31% | 1.79% | 70.23% | 11.53% | 9.30% | 11.09% | 13.84% | 4.79% |
| Kashgar | 0.08% | 4.57% | 2.63% | 2.51% | 1.19% | 65.63% | 17.09% | 6.30% | 7.49% | 19.60% | 7.20% |
| Minqin | 0.11% | 2.97% | 2.33% | 6.13% | 2.33% | 65.70% | 12.20% | 8.23% | 10.56% | 18.33% | 5.30% |
| Xi'an | 0.60% | 4.74% | 9.62% | 10.08% | 6.56% | 60.75% | 3.37% | 4.28% | 10.84% | 13.45% | 14.37% |
| Beijing | 0.69% | 5.91% | 7.67% | 10.65% | 6.08% | 62.86% | 2.85% | 3.29% | 9.38% | 13.49% | 13.59% |
| Nanjing | 0.96% | 6.26% | 15.13% | 13.79% | 9.68% | 46.94% | 3.14% | 4.10% | 13.78% | 16.93% | 21.39% |
| Shanghai | 1.30% | 5.14% | 10.78% | 17.68% | 10.33% | 46.62% | 3.87% | 4.28% | 14.61% | 21.55% | 15.92% |
| Harbin | 0.97% | 5.43% | 12.45% | 15.36% | 10.80% | 50.50% | 2.21% | 2.26% | 13.07% | 17.57% | 17.88% |
| Hefei | 0.80% | 3.33% | 11.51% | 14.86% | 10.34% | 49.79% | 3.91% | 5.46% | 15.80% | 18.77% | 14.84% |
| Songshan | 0.59% | 8.14% | 6.31% | 12.09% | 6.63% | 59.65% | 3.10% | 3.50% | 10.13% | 15.18% | 14.45% |
| Chengdu | 0.58% | 2.72% | 11.90% | 22.54% | 9.34% | 42.10% | 3.65% | 7.17% | 16.51% | 26.19% | 14.62% |
| Zhoushan | 0.33% | 3.55% | 6.84% | 17.06% | 14.86% | 42.82% | 5.83% | 8.70% | 23.56% | 22.89% | 10.40% |
| Guangzhou | 0.64% | 2.84% | 8.18% | 23.80% | 18.26% | 31.54% | 4.28% | 10.46% | 28.72% | 28.08% | 11.02% |
| Haikou | 0.83% | 2.32% | 9.89% | 22.03% | 14.42% | 33.29% | 6.90% | 10.32% | 24.74% | 28.93% | 12.21% |
| Sanya | 0.55% | 0.32% | 14.27% | 24.78% | 9.46% | 37.41% | 3.21% | 10.00% | 19.45% | 27.99% | 14.59% |

Figure 6. Why only imaginary for fine and only 670? Fitting statistics is a necessary (but not sufficient) metric to justify that the method works. Please consider showing at least a real part at 670 too, but I would be more convinced to see all of them, all wl and fine/coarse. At least show minimal set, following the structure of Table s2 fine/coarse real/imaginary and 440 too, since they all referenced to have different level of uncertainty. Also, please, mention that it is fine mode in figure caption.

**Response:** We add the closure figure for the real and imaginary parts of the CRI at 440 nm. Because the spectral

changes in the real part are ignored in the CRI inversion, the optical closure at other wavelengths have a similar pattern (We don't show that in the manuscript). The Figure is as follows and its description has been mentioned in the previous response.

[Figure]

**Figure 8**. Data pair frequency of instantaneous imaginary parts of the complex refractive index at 675 nm ($k_f$), 440 nm ($k_{f,440}$), and real part at 440 nm ($n_f$) which are sorted according to ordered pairs (Retrieved, Estimated) in 0.0005 and 0.001 intervals for imaginary and real parts, respectively. "Retrieved" represents sub-component of CRI from the optical-physical retrievals, and "Estimated" is estimated by retrieved chemical components. The color represents the number of cases (color bar), and the solid black line shows the 1:1 line.

For the coarse mode, only the real part of CRI ($n_c$) is used in the inversion because the imaginary part of the CRI for both $AW_c$ and SC are zero. Figure R2 shows the comparison of $n_c$ between retrieved and estimated from chemical components. The residual is tiny when the $n_c$ is less than 1.534. To the left of the 1:1 line, when $n_c$ equals 1.534 (real part of DU), there is a set of points with a large error. That is because the relative humidity is lower than 40% leading to the $AW_c$ is close to 0.0. Hence, the real part of the coarse mode gets the minimum of 1.534. In contrast, the real part of the CRI for the DU component is increased up to the mean value of the retrieved refractive index at all available wavelengths which is higher than the mean value of DU's $n_c$. The reason for doing that is that DU is a mixture. Even so, when relative humidity is high, some underestimates still occur. Fortunately, 73% of the points are not influenced by the above factors, with the mean error of 0.01. The uncertainty of the retrieved CRI also causes the large residuals.

[Figure]

Figure R2. Data pair frequency of instantaneous imaginary parts of the complex refractive index at 440 nm ($n_c$) which are sorted according to ordered pairs (Retrieved, Estimated) in 0.001 intervals. The color represents the number of cases (color bar), and the solid black line shows the 1:1 line.

Figure 7. Is it possible to re-plot or at least re-paint so the color scheme will be the same as in figs 5 and 6?
**Response:** We re-plot this figure to match the color scheme of the other figures.

[Figure]

**Figure 9**. The mass concentrations of aerosol components in four seasons (winter, spring, summer and autumn). For the box-whisker plot, the mean value is indicated by a plus sign (+), and the median value by a short line inside the box (−). The top and bottom edges of each box represent the top and bottom quartiles (Q3 and Q1), and the corresponding whiskers are the outliers (Q3+1.5IQR and Q1-1.5IQR, IQR is interquartile range).

Table 2. Please specify what "No" in the head of the table means. If this is a number of observations, then why for quite different observation periods the numbers are so close? Was observation data filtered? Please, clarify.

**Response:** "No" is the meteorological station number. We explain it at the bottom of table 2.

Table s2 and s1. They are not a part of a publication provided (.pdf) and I don't see much reason why. I mean they are rather small and multiply referenced from the text, what is the reason to have one external half-page containing them?

**Response:** These two tables are referenced from other references to provide model input errors. For clarity, we added a description of them to the supplementary (S1).

**Technical comments:**

Figure 3. Size distribution axis. I believe it is μm3 *μm-2

**Response:** Thank you for pointing this out. We revised figure as follows:

[Figure]

**Figure 6**. The fine and coarse-mode volume size distribution, complex refractive index and aerosol components of three typical aerosol models (WS: water soluble, BB: biomass burning, DU: dust).

Line 27-28: "… retrieval of the chemical composition …"
**Response:** Following the comment of another reviewer, we have deleted this sentence.

Line 49: "… simultaneously retrieved …"
**Response:** We revised this sentence as follows:
Line 52: "Zhang et al. (2018) simultaneously retrieved the WSOM and WIOM components but ignored the error in the refractive index introduced by the aerosol volume averaging method applied to the multicomponent liquid system."

Line 56: "… to solve for the refractive index in a multicomponent liquid system"
**Response:** We revised this sentence as follows:
Line 59: "hygroscopicity is introduced to solve for the refractive index in a multicomponent liquid system."

Line 57: "… in the algorithm …"
**Response:** We revised this sentence as follows:
Line 60: "The results are used in the algorithm to retrieve aerosol components … "

Line 70: "… radiation to determine the columnar water vapor"
**Response:** We revised this sentence as follows:
Line 73: "All bands provide both radiation and polarization measurements, except the 936 nm band which only measures radiation to determine the columnar water vapor."

Line 93-94: Probably author meant "To avoid observation uncertainties, only data from manned weather stations that are maintained regularly were used."

**Response:** We modified this sentence into the form suggested by the reviewer.

Line 185: "… by the different aerosol size distribution …"

**Response:** We revised this sentence as follows:

Line 243: "Each type is described by the different aerosol size distribution and refractive index parameters derived from Zhang et al. (2017)."

Line 246: "On the one hand"

**Response:** We revised this sentence as follows:

Line 310: "On the one hand, the imaginary part of the refractive index of BC is significantly higher than for the other components due to its strong absorption."

Line 247: "… than for the other components …"

**Response:** We revised this sentence as follows:

Line 310: "On the one hand, the imaginary part of the refractive index of BC is significantly higher than for the other components due to its strong absorption."

Line 252: "… data points are below …"

**Response:** We revised this sentence as follows:

Line 316: "The mean relative error (RE) is 27.1%, and 62.8% of the data points are below the average relative error line."

Line 260: "… high in spring …"
Line 260: "… to the dust transport from the northwest China …"

**Response:** We revised this sentence as follows:

Line 342-343: "Low concentrations of BC in the other seasons are mainly due to the influence of frequent dust events in the spring and high aerosol hygroscopic growth in the summer."

Line 261: "… seasons indicates that the aerosol at some sites"

**Response:** We revised this sentence as follows:

Line 337: "The difference between the upper and lower quartiles of $AW_f$ in the summer is larger than in other seasons indicating that in the summer the aerosol at some sites has a low hygroscopicity."

Line 262: "… due to the observational errors …"

**Response:** We revised this sentence as follows:

Line 373: "The unusually high values at Shanghai in 2016 may be due to the observational errors."

Line 304: "… changes in the meteorological conditions …"

**Response:** We revised this sentence as follows:

Line 385: " Coarse mode aerosols usually derive from natural sources, and their variations can be associated with changes in the meteorological conditions."

---

## Author Response (AR2)

*General comments:*

*This paper describes a method that estimates the aerosol components by calculating the refractive index of a aerosol mixture. Then the algorithm was applied to ground-based remote sensing measurements to retrieve the aerosol components in China. The information of aerosol component is important for the understanding of climate change, air quality, the interaction between aerosols and cloud, chemical transport model estimation, etc. Meanwhile, the concentration of aerosol components in the atmospheric column are quite difficult to measured. Therefore, the efforts on retrieval of aerosol component in this study are commendable and the work is meaningful. However, I have some comments on the current manuscript.*

*Major comments:*

*1. I think the authors should highlight the improvements of aerosol component retrieval in their study rather than some results that are well-known in many previous studies in the abstract, such as "the atmospheric columnar DU component is dominant in the northern region of China, whereas the AW is higher in the southern coastal region". Because the title of "improved inversion of aerosol components in the atmospheric column from remote sensing data" emphasizes the new development of algorithm. I also suggest to show some comparisons of aerosol component retrievals between the improved algorithm and the previous algorithm. As Referee #3 mentioned in "Major comments 3: a comparison to a previous method could be presented, if, of course, such comparison could be done. For example, comparison with OM from Zhang 2018 could be performed to illustrate improvements (if any)"*

**Response:** Thank you for this comment. We added the improvements of this algorithm and some new findings in the abstract.

Line 10-25: "**Abstract.** Knowledge on the composition of atmospheric aerosols is important for reducing the uncertainty of climate assessment. In this study, an improved algorithm is developed for the retrieval of atmospheric columnar aerosol components from optical remote sensing data. This is achieved by using the complex refractive index (CRI) of a multicomponent liquid system in the forward model and minimizing the differences with the observations. The aerosol components in this algorithm comprise five species, combining eight sub-components including black carbon (BC), water-soluble (WSOM) and water-insoluble organic matter (WIOM), ammonium nitrate (AN), sodium chloride (SC), dust-like (DU), and aerosol water content in the fine and coarse modes (AW$_f$ and AW$_c$). The calculation of the CRI in the multicomponent liquid system allows to separate the water-soluble components (AN, WSOM and AW$_f$) in the fine mode and the SC and AW$_c$ in the coarse mode. The uncertainty in the retrieval results is analyzed based on the simulation of typical models, showing that the complex refractive index obtained from instantaneous optical-physical inversion compares well with that obtained from chemical estimation. The algorithm was used to retrieve the columnar aerosol components over China using the ground-based remote sensing measurements from the Sun-sky radiometer Observation NETwork (SONET) in the period from 2010 to 2016. The results were used to analyze the regional distribution and interannual variation. The analysis shows that the atmospheric columnar DU component is dominant in the northern region of China, whereas the AW is higher in the southern coastal region. The SC component retrieved over the desert in northwest China originates from a paleo-marine source. The AN significantly decreased from 2011 to 2016, by 21.9 mg m$^{-2}$, which is inseparable from China's environmental control policies. "

We add the comparison with Zhang et al. (2018) in the supplementary S3:

**"S3. The comparison of aerosol components**

We have made the comparison of the aerosol components retrieved with the new algorithm presented here, with those from Zhang et al (2018). The number of retrievals in this study is less than that in Zhang et al. (2018). There are three reasons: (1) The input data is more rigorously filtered (Li et al., 2018); (2) the residuals are increased using the new algorithm; (3) stricter residual constraints are used. From these, we can obtain more reasonable inversion aerosol components. Figure S1 shows the comparison of aerosol components (OM, BC and AN) in the fine mode in atmospheric column from this study and those from Zhang et al., 2018 with reference $PM_1$ composition data which were measured by a High-Resolution Aerosol Mass Spectrometer at ground level. We use the boundary layer height of lidar (obtained from Zhang et al. 2018) to calculate the concentration of the atmospheric column to the near surface. The results show that OM components from the improved algorithm are not better than from Zhang et al. (2018). Black carbon is closer to the identity line although the correlation coefficient is slightly smaller than in 2018. For AN, a water-soluble inorganic salt, the new algorithm shows a good effect. The slope with ground observations changes from negative to positive. In our opinion, such a comparison is not sufficient due to the various vertical distribution of aerosol components. In future studies, we will make a more detailed and comprehensive comparison.

This comparison does not show the comprehensive advantages of the new algorithm. Although the algorithm in this paper has been improved, the basic assumption (e.g. Nonhygroscopic assumption of OM mixture) is not different from the paper in 2018. The current algorithm can easily add more kinds of hygroscopic components without obtaining the single component hygroscopic formula (A polynomial in water activity and solution concentration in paper of 2018 Eqs (5) & (6)) to better solve the problem of OM mixture."

But we may need more work to understand the properties of OM. We are publishing this work in the hope that more scholars will join us to improve the inversion of aerosol components based on our flexibility algorithm.

[Figure]

Figure S1. The comparison of aerosol components (OM, BC and AN) between this study and Zhang et al., 2018.

Li, Z., Xu, H., Li, K. T., Li, D. H., Xie, Y. S., Li, L., et al.: Comprehensive study of optical, physical, chemical, and radiative properties of total columnar atmospheric aerosols over China: An overview of Sun–Sky Radiometer Observation Network (SONET) measurements. Bulletin of the American Meteorological Society, 99(4), 739–755, 2018.

*2. Line 217-223: You mentioned that "The improved algorithm described here is more suitable for the calculation of the properties of a mixture of multiple water-soluble components …" and the previous algorithm had some limits. Could you provide the comparisons of aerosol component retrieval derived by these two algorithms? For example, aerosol water fraction. Because I wonder if these two approaches can obtain similar results for aerosol water fraction that is considered in both of two algorithms. The aerosol water fraction should have a good agreement between the improved algorithm and previous algorithm (Zhang, 2018).*

**Response:** We add the comparison with $AW_f$ from Zhang et al. (2018) in the supplementary S3: "The daily volume fraction of $AW_f$ from the algorithm of 2020 and 2018 is present in figure R2. The volume fraction of $AW_f$ obtained by the two algorithms is consistent with the change of relative humidity. $AW_f$ from the algorithm of 2020 is slightly higher than that of 2018. The new algorithm increases the low $AW_f$ when the RH is more than 40%, obtaining more reasonable results. "

[Figure]

Figure S2. The RH and daily volume fraction of $AW_f$ from the algorithm of 2020 and 2018.

*3. Could you provide a table to show the statistics of fitting for each aerosol component in Figure 5?*

**Response:** The statistics of fitting for aerosol components in figure 5 is listed in table R1.

Table R1. The slope, intercept and $R^2$ for aerosol components in figure 5.

|  | BC | WIOM | WSOM | AN | $AW_f$ | DU | SC | $AW_c$ |
|---|---|---|---|---|---|---|---|---|
| **Slope** | 1.27 | 0.84 | 0.84 | 0.93 | 0.90 | 0.97 | 1.00 | 0.95 |
| **Intercept** | 0.00 | 0.02 | 0.07 | 0.00 | 0.03 | 0.03 | 0.01 | 0.00 |
| **$R^2$** | 0.98 | 0.97 | 0.90 | 0.98 | 0.86 | 0.99 | 0.99 | 0.99 |

*4. I also read the paper (Zhang et al., 2018), which is you cited and mentioned in the current manuscript. The values of aerosol component density used in Zhang et al. (2018) (Table 1) are quite different to that used in the current manuscript (Table 2), but with same values of complex refractive index. Why? I suggest to use same values of density in the current manuscript as that in Zhang et al. (2018), because more uncertainty could be induced from the density. For example, WIOM and WSOM density in Zhang et al. (2018) is 1.0, whereas it is 1.547 in current manuscript. The uncertainty could be up to more than 50%. In this case, the results in Figure 8,9 and 10 may have some changes.*

**Response:** Thanks for pointing this out. The OM density used in the model (e.g. WRF-Chem) and

observation (e.g. Zhang et al., 1993) were 1.0, so this value was used in the paper in 2018. But in later studies of component remote sensing larger values were used (e.g. Xie et al., 2017, van Beelen et al., 2014). In the current study we followed this and used van Beelen et al. (2014). They use weighted density for OM from 20%wt levoglucosan, 40%wt succinic acid, and 40% Suwannee River reference fulvic acid. The densities for other aerosol components were modified accordingly. We quoted the literature incorrectly in the manuscript, which has been modified as follows:

Line 137-138: "In the current study the effective density of aerosol components is used from a widely cited study by van Beelen et al. (2014)."

And Table 2:

**Table 2**. Growth factor derived hygroscopicity parameter ($\kappa$), complex refractive indexes ($m = n - ik$) at four wavelengths and effective density ($\rho$) of model components. Real and imaginary parts at four standard AERONET aerosol product wavelengths (440, 675, 870 and 1020 nm) are considered.

| Component | | Growth factor derived $\kappa$ | Real Part | | | | Imaginary Part | | $\rho$ |
|---|---|---|---|---|---|---|---|---|---|
| | | | $n_{440}$ | $n_{675}$ | $n_{870}$ | $n_{1020}$ | $k_{440}$ | $k_{675\sim1020}$ | (g cm$^{-3}$) |
| OM | WIOM | 0.000 | 1.530[c] | 1.530 | 1.530 | 1.530 | 0.035[d] | 0.001 | 1.547[i] |
| | WSOM | 0.000[a] | 1.530[c] | 1.530 | 1.530 | 1.530 | 0.006[d] | 0.000 | |
| AN | | 0.547[b] | 1.559[e] | 1.553 | 1.550 | 1.548 | 0.000[e] | 0.000 | 1.760[i] |
| BC | | 0.000 | 1.950[f] | 1.950 | 1.950 | 1.950 | 0.790[f] | 0.790 | 1.800[i] |
| AW | | 0.000 | 1.337[e] | 1.332 | 1.330 | 1.328 | 0.000[g] | 0.000 | 1.000[i] |
| DU | | 0.000 | 1.534[g] | 1.534 | 1.534 | 1.534 | 0.002[h] | 0.001 | 2.650[i] |
| SC | | 1.120[a] | 1.562[h] | 1.541 | 1.534 | 1.530 | 0.000[i] | 0.000 | 2.165[i] |

[a] Petters and Kreidenweis, 2007; [b] Kreidenweis et al., 2008; [c] Sun et al., 2007; [d] Chen and Bond, 2010; [e] Schuster et al., 2005; [f] Bond and Bergstrom, 2006; [g] Koven and Fung, 2006; [h] Toon et al., 1976; [i] van Beelen et al., 2014.

van Beelen, A. J., Roelofs, G. J. H., Hasekamp, O. P., Henzing, J. S., and Rockmann, T.: Estimation of aerosol water and chemical composition from AERONET Sun-sky radiometer measurements at Cabauw, the Netherlands, Atmos. Chem. Phys., 14(12), 5969–5987, doi:10.5194/acp-14-5969-2014, 2014.

Xie, Y., Li, Z., Zhang, Y.X., Zhang, Y., Li, D. H., Li, K. T. Xu, H., Zhang, Y., Wang, Y. Q., Chen, X. F., Schauer, J. J., Bergin, M.: Estimation of atmospheric aerosol composition from ground-based remote sensing measurements of Sun-sky radiometer, Journal of Geophysical Research Atmospheres, doi: 10.1002/2016JD025839, 2017.

Zhang, X.Q., Mcmurry, P.H., Hering, S.V., Casuccio, G.S.: Mixing characteristics and water content of submicron aerosols measured in Los Angeles and at the Grand Canyon. Atmos. Environ. 27A (10), 1593–1607, 1993.

*5. Could you provide some validation of aerosol component retrievals? For example, the validation for black carbon concentration with in situ measurements.*

**Response:** Our retrievals provide the aerosol composition integrated over the whole atmospheric column, which is usually quite different from that near the surface measured by in situ measurements. This is due to vertical variations, the occurrence of disconnected layers, height dependent chemistry etc. Therefore, a direct comparison between column integrated and ground-based data is not meaningful, unless corrections are made based on comprehensive understanding of atmospheric processes (physical models). Nevertheless, as shown in Figure S1, we compared our results with the ground-based observations in $PM_1$. This is the only observation we have of chemical components. However, it is not enough to assess the quality of our results. Perhaps we need to carry out flight experiments over China to better verify the aerosol components in atmospheric column.

In addition, lacking suitable reference data, the discussion in Section 4 shows the spatial distribution, seasonality and interannual variations with (figs 8-11) which show trends as may be expected.

Minor comments:

1. The definition of "aerosol water" is not appropriate and it cannot describe exactly the aerosol component. I suggest to use "aerosol water content" in the paper.

**Response:** Thanks for this comment. We corrected the manuscript overall.

2. Line 23: "aerosol particles scatter and absorb solar radiation" It is imprecise.

**Response:** This sentence has been removed.

3. Please reword the sentence of "the detail of information depends on the technique used" (Line 27)

**Response:** This sentence has been reworded as follows:

Line 30-31: "Each technique provides information on the aerosol composition which may differ in content and detail."

4. Line 217: "In that algorithm" should be "In previous algorithm"

**Response:** We revise to: "in the previous algorithm".

5. Line 234: "in good agreement" should be "in a good agreement"

**Response:** We checked the Webster dictionary and decided to rephrase to

Line 235-236: "The volume fractions of the retrieved aerosol components reproduce the input values reasonably well."

6. Line 230-231: "…the volume fraction of BC was constrained between 0 to 3.0%…". Why?

**Response:** Because Zhang et al. (2012) analyzed the BC component in $PM_{10}$ over China and found that its mass concentration fraction account for about 3.5%. Because BC density is relatively high, we estimate the volume fraction at 3% to perform the tests.

[revised manuscript text omitted]

**S1. Estimation of CRIs for the fine and coarse modes**

In order to separate the complex refractive index (CRI) for different modes, first the volume size distribution (VSD) needs to be separated into complete log-normal functions following the VSD breakdown method. The multi-modal log-normal distributions fits the AERONET-retrieved VSD by the following formula:

$$\frac{dV(r)}{d\ln r} = \sum_{i=1,m} \frac{C_i}{\sqrt{2\pi}|\ln\sigma_i|} exp\left[-\frac{1}{2}\left(\frac{\ln r - \ln r_i}{\ln\sigma_i}\right)^2\right], \text{ m=1, 2, ...,} \tag{S1}$$

where $dV/d\ln r$ (in unit of $\mu m^3/\mu m^2$) is the volume particle size distribution, $C_i$ ($\mu m^3/\mu m^2$), $r_i$ ($\mu m$), and $\ln \sigma_i$ are the volume modal concentration, median radius, and standard deviation of $\ln r_i$ for each log-normal mode, respectively. Based on the separated VSD for the fine and coarse mode, a limited-memory optimization algorithm is employed to retrieve the CRIs. The real part (n) of sub-CRIs is spectrally independent, and the imaginary part (k) of sub-CRIs varies with wavelength:

$$n_{f/c}(\lambda) = n_{f/c} \tag{S2}$$

$$k_{f/c}(\lambda) = \begin{cases} k_{f/c,440} \\ k_{f/c} \end{cases}$$

where $\lambda$ denotes the standard wavelengths of AERONET products and "*f*" and "*c*" represent the fine and coarse modes, respectively. The details of the CRIs separating process are presented in Zhang et al. (2017).

In the study of Zhang et al. (2017), the uncertainties in the estimated complex refractive indices of the fine and coarse modes were evaluated using the three typical aerosol models presented in table S1. The typical uncertainties in the retrieved complex refractive indices of fine and coarse modes for these models are listed in the table 2. The total uncertainty (TU) is calculated by error propagation using the

formula:

$$TU = \sqrt{x_{\Delta\tau}^2 + x_{\Delta\omega}^2 + x_{\Delta VSD}^2} \qquad (S3)$$

where $x$ represents the uncertainty of the sub-CRIs for each aerosol type. The biases of the input parameter (aerosol optical depth ($\tau$), single scattering albedo ($\omega$) and VSD) uncertainties are set to 0.01, -0.03 and 15%-35% in the WS, BB and DU aerosol models, respectively. The uncertainty in the relative humidity is twice the observational error given by the World Meteorological Organization (WMO, 2008).

**Table S1**. Typical aerosol models (WS: Water-soluble, BB: Biomass burning, DU: Dust) parameters and relative humidity.

| Type | $r_1$ | $r_2$ | $\sigma_1$ | $\sigma_2$ | $C_1/C_2$ | $n_f$ | $k_{f,440}$ | $k_f$ | $n_c$ | $k_{c,440}$ | $k_c$ | RH |
|------|-------|-------|-----------|-----------|-----------|-------|-------------|-------|-------|-------------|-------|-----|
| **WS** | 0.118 | 1.17 | 0.6 | 0.6 | 2 | 1.45 | 0.0035 | 0.0035 | 1.53 | 0.008 | 0.008 | 70% |
| **BB** | 0.132 | 4.50 | 0.4 | 0.6 | 4 | 1.52 | 0.025 | 0.025 | 1.53 | 0.008 | 0.008 | 55% |
| **DU** | 0.100 | 3.40 | 0.6 | 0.8 | 0.066 | 1.53 | 0.008 | 0.008 | 1.53 | 0.008 | 0.008 | 35% |

**Table S2.** Typical uncertainties ($\pm\Delta$) of the estimated complex refractive indices of the fine and coarse modes and relative humidity.

| Type | $\Delta n_f$ | $\Delta k_{f,440}$ | $\Delta k_f$ | $\Delta n_c$ | $\Delta k_{c,440}$ | $\Delta k_c$ | $\Delta RH$ |
|------|-------------|--------------------|--------------|--------------|--------------------|--------------|-------------|
| WS | 0.0197 | 0.0015 | 0.0004 | 0.0667 | 0.0036 | 0.0029 | 10% |
| BB | 0.0270 | 0.0039 | 0.0030 | 0.0500 | 0.0016 | 0.0016 | 10% |
| DU | 0.0780 | 0.0036 | 0.0040 | 0.0447 | 0.0016 | 0.0036 | 10% |

**S2. The averaged mass fraction of aerosol components**

**Table S3. The averaged mass fraction of aerosol components at SONET sites shown in figure 7.**

| Site | Fine mode | | | | | Coarse mode | | | AW | IS | OM |
|------|-----------|------|------|------|------|-------------|------|------|------|------|------|
| | BC | WIOM | WSOM | AN | AWf | DU | SC | AWc | | | |
| Lhasa | 0.43% | 7.85% | 6.47% | 1.89% | 2.01% | 81.34% | 0.00% | 0.00% | 2.01% | 1.89% | 14.33% |
| Zhangye | 0.05% | 2.43% | 2.36% | 2.31% | 1.79% | 70.23% | 11.53% | 9.30% | 11.09% | 13.84% | 4.79% |
| Kashgar | 0.08% | 4.57% | 2.63% | 2.51% | 1.19% | 65.63% | 17.09% | 6.30% | 7.49% | 19.60% | 7.20% |
| Minqin | 0.11% | 2.97% | 2.33% | 6.13% | 2.33% | 65.70% | 12.20% | 8.23% | 10.56% | 18.33% | 5.30% |
| Xi'an | 0.60% | 4.74% | 9.62% | 10.08% | 6.56% | 60.75% | 3.37% | 4.28% | 10.84% | 13.45% | 14.37% |
| Beijing | 0.69% | 5.91% | 7.67% | 10.65% | 6.08% | 62.86% | 2.85% | 3.29% | 9.38% | 13.49% | 13.59% |
| Nanjing | 0.96% | 6.26% | 15.13% | 13.79% | 9.68% | 46.94% | 3.14% | 4.10% | 13.78% | 16.93% | 21.39% |
| Shanghai | 1.30% | 5.14% | 10.78% | 17.68% | 10.33% | 46.62% | 3.87% | 4.28% | 14.61% | 21.55% | 15.92% |
| Harbin | 0.97% | 5.43% | 12.45% | 15.36% | 10.80% | 50.50% | 2.21% | 2.26% | 13.07% | 17.57% | 17.88% |
| Hefei | 0.80% | 3.33% | 11.51% | 14.86% | 10.34% | 49.79% | 3.91% | 5.46% | 15.80% | 18.77% | 14.84% |
| Songshan | 0.59% | 8.14% | 6.31% | 12.09% | 6.63% | 59.65% | 3.10% | 3.50% | 10.13% | 15.18% | 14.45% |
| Chengdu | 0.58% | 2.72% | 11.90% | 22.54% | 9.34% | 42.10% | 3.65% | 7.17% | 16.51% | 26.19% | 14.62% |
| Zhoushan | 0.33% | 3.55% | 6.84% | 17.06% | 14.86% | 42.82% | 5.83% | 8.70% | 23.56% | 22.89% | 10.40% |
| Guangzhou | 0.64% | 2.84% | 8.18% | 23.80% | 18.26% | 31.54% | 4.28% | 10.46% | 28.72% | 28.08% | 11.02% |
| Haikou | 0.83% | 2.32% | 9.89% | 22.03% | 14.42% | 33.29% | 6.90% | 10.32% | 24.74% | 28.93% | 12.21% |
| Sanya | 0.55% | 0.32% | 14.27% | 24.78% | 9.46% | 37.41% | 3.21% | 10.00% | 19.45% | 27.99% | 14.59% |

**S3. The comparison of aerosol components**

We have made the comparison of the aerosol components retrieved with the new algorithm presented here, with those from Zhang et al (2018). The number of retrievals in this study is less than that in Zhang et al. (2018). There are three reasons: (1) The input data is more rigorously filtered (Li et al., 2017); (2) the residuals are increased using the new algorithm; (3) stricter residual constraints are used. From these, we can obtain more reasonable inversion aerosol components. Figure S1 shows the comparison of aerosol components (OM, BC and AN) in the fine mode in atmospheric column from this study and those from Zhang et al., 2018 with reference $PM_1$ composition data which were measured by a High-Resolution Aerosol Mass Spectrometer at ground level. We use the boundary layer height of lidar (obtained from Zhang et al. 2018) to calculate the concentration of the atmospheric column to the near surface. The results show that OM components from the improved algorithm are not better than from Zhang et al. (2018). Black carbon is closer to the identity line although the correlation coefficient is slightly smaller than in 2018. For AN, a water-soluble inorganic salt, the new algorithm shows a good effect. The slope with ground observations changes from negative to positive.

In our opinion, such a comparison is not sufficient due to the various vertical distribution of aerosol components. In future studies, we will make a more detailed and comprehensive comparison.

This comparison does not show the comprehensive advantages of the new algorithm. Although the algorithm in this paper has been improved, the basic assumption (e.g. Nonhygroscopic assumption of OM mixture) is not different from the paper in 2018. The current algorithm can easily add more kinds of hygroscopic components without obtaining the single component hygroscopic formula (A polynomial in water activity and solution concentration in paper of 2018 Eqs (5) & (6)) to better solve the problem of OM mixture.

[Figure]

**Figure S1**. The comparison of aerosol components (OM, BC and AN) between this study and Zhang et al., 2018.

The daily volume fraction of $AW_f$ from the algorithm of 2020 and 2018 is present in figure S2. The volume fraction of $AW_f$ obtained by the two algorithms is consistent with the change of relative humidity. $AW_f$ from the algorithm of 2020 is slightly higher than that of 2018. The new algorithm increases the low $AW_f$ when the RH is more than 40%, obtaining more reasonable results.

[Figure]

**Figure S2**. The RH and daily volume fraction of $AW_f$ from the algorithm of 2020 and 2018.